# A natural biogenic nanozyme for scavenging superoxide radicals

Long Ma[1,2], Jia-Jia Zheng[3], Ning Zhou[1], Ruofei Zhang[1], Long Fang[1,4], Yili Yang[5], Xingfa Gao [3], Chunying Chen [6], Xiyun Yan [1,2,7,8] ✉ & Kelong Fan [1,2,7,8] ✉

Biominerals, the inorganic minerals of organisms, are known mainly for their physical property-related functions in modern living organisms. Our recent discovery of the enzyme-like activities of nanomaterials, coined as nanozyme, inspires the hypothesis that nano-biominerals might function as enzyme-like catalyzers in cells. Here we report that the iron cores of biogenic ferritins act as natural nanozymes to scavenge superoxide radicals. Through analyzing eighteen representative ferritins from three living kingdoms, we find that the iron core of prokaryote ferritin possesses higher superoxide-diminishing activity than that of eukaryotes. Further investigation reveals that the differences in catalytic capability result from the iron/phosphate ratio changes in the iron core, which is mainly determined by the structures of ferritins. The phosphate in the iron core switches the iron core from single crystalline to amorphous iron phosphate-like structure, resulting in decreased affinity to the hydrogen proton of the ferrihydrite-like core that facilitates its reaction with superoxide in a manner different from that of ferric ions. Furthermore, overexpression of ferritins with high superoxide-diminishing activities in *E. coli* increases the resistance to superoxide, whereas bacterioferritin knockout or human ferritin knock-in diminishes free radical tolerance, highlighting the physiological antioxidant role of this type of nanozymes.

Inorganic minerals naturally occurring as homogeneous solids with a definite chemical composition, are widely distributed on the earth. Until now, there are almost 5800 different kinds of minerals recorded by the International Mineralogical Association (IMA)[1]. According to Robert M. Hazen, the mineral evolution has experienced 10 partially overlapping stages, from the occurrence of primary chondritic minerals to the phanerozoic biomineralization[2]. Notably, nearly two-thirds of all the natural minerals resulted from the transformation of the earth by living organisms, especially in the period after the "Great Oxidant Event"[3], indicating the close relationship between mineral generation and life evolution. Indeed, the minerals not only are believed to participate in the origin of life but also have become integrated components of organisms in most, if not all the living kingdoms[4]. In the prebiotic time, the enzyme architecture was considered too complicated to be available. Inorganic minerals were supposed to catalyze the synthesis and polymerization of the initial

[1]CAS Engineering Laboratory for Nanozyme, Key Laboratory of Biomacromolecules (CAS), CAS Center for Excellence in Biomacromolecules, Institute of Biophysics, Chinese Academy of Sciences, Beijing 100101, China. [2]University of Chinese Academy of Sciences, Chinese Academy of Sciences, Beijing 100408, China. [3]Laboratory of Theoretical and Computational Nanoscience, National Center for Nanoscience and Technology of China, Beijing 100190, China. [4]Savaid Medical School, University of Chinese Academy of Sciences, Beijing 100049, China. [5]China Regional Research Centre, International Centre for Genetic Engineering and Biotechnology, Taizhou, Jiangsu 225316, China. [6]CAS Key Laboratory for Biomedical Effects of Nanomaterials and Nanosafety and CAS Center for Excellence in Nanoscience, National Center for Nanoscience and Technology, Beijing 100190, China. [7]Nanozyme Medical Center, School of Basic Medical Sciences, Zhengzhou University, Zhengzhou, Henan 450052, China. [8]Nanozyme Laboratory in Zhongyuan, Zhengzhou, Henan 451163, China. ✉e-mail: yanxy@ibp.ac.cn; fankelong@ibp.ac.cn

biomolecules[5]. The minerals could catalyze the primary organic synthesis, like the carbon fixation under the primordial environment[6,7], and realize the chiral molecule selection on the chiral surface of natural minerals, like the calcite[8]. In addition to the small molecule synthesis and concentration, polymerization of the peptides and RNA molecules could also be catalyzed by the clay minerals such as the layered double hydroxide mineral[9,10]. In modern living organisms, there are considerable amounts of biominerals in almost all the species, such as the carbonate and magnetite in bacteria[11], calcium oxalate in plants[12] and bones in animals. However, unlike their catalytic roles in the prebiotic period, the biominerals play mainly physical support and protective roles, such as the supporting role of silicate in diatom and bones in animals[13].

The catalytic role of the biomineralized materials was little discussed until the discovery of the enzyme-like activity of the inorganic nanomaterials[14]. These nanozymes have since been widely studied in various areas including environmental science, the pharmaceutical industry, and so on[15,16]. In addition to their intrinsic physical and chemical properties like magnetism and fluorescence[17,18], the nanozymes have catalytic activities and reaction kinetics similar to the natural enzymes[19,20]. Currently, a variety of nanozymes have been developed from the original magnetic particles to thousands of different nanomaterials and even the single-atom nanozyme that could match the natural horseradish peroxidase[21]. These unique properties make nanozyme a preferred enzyme mimic for applications in environmental protection, agriculture, disease diagnosis and treatment[22].

Interestingly, there have been a few published reports on natural nanozymes from biominerals, including the magnetosome in magnetic bacteria, bio-interface mineral nanozyme in the fungus and horse spleen ferritin[23–25], which exhibit peroxidase-like activities in addition to being natural constituents. Furthermore, it was shown that the magnetosome and fungal bio-interface mineral nanozyme played an antioxidant role in protecting their host from the superoxide or hydroxyl radical, indicating that these nanozymes might have an important physiological function. Given the diverse and wide distribution of biominerals, it is conceivable that natural nanozymes are present and functional in multiple species of living organisms. Of note, iron storage protein ferritin exists as a natural nano-structure in almost all living organisms including bacteria, archaea, and eukaryotes[26]. Despite significant differences in their sequences, ferritins from all species store iron in the form of the inorganic iron core, a classical type of inorganic nanomaterials in vivo.

In the present study, we examine 18 ferritins from bacteria, archaea, and eukaryotes for their nanozyme activities and decipher the underlying mechanisms. It is found that, compared to eukaryote ferritins, the prokaryote ferritins exhibit higher superoxide dismutase (SOD)-like activity. Our data demonstrate that the differences in SOD-like activity of these ferritins result from the changes of the iron/phosphate ratio in the iron cores. The archaea and bacteria ferritins possess higher phosphate content in relative to human ferritin, which is determined by their amino acid sequences and structures. Furthermore, extended X-ray absorption fine structure (EXAFS) and density functional theory (DFT) analyses show that the phosphate inner the iron core, not the surface one, changes the hydrogen proton binding ability of the iron core, which affects the reaction ability with the superoxide. More importantly, the ferritins with high SOD-like activities can play an antioxidant role in the psychological condition. These results demonstrate the biogenic ferritin iron cores from different species have distinct abilities to scavenge superoxide radicals. The structures of the ferritins determine the composition of the iron cores that in turn decide the SOD-like activity, illustrating that the enzyme-like activity of the natural nano-biomineral is structure-dependent and evolutionary conserved, and may play an important role in maintaining the homeostasis of living organisms.

## Results and discussion

### Biogenic iron cores of ferritins from distinct organisms exhibiting different SOD-like activities

Based on their structures, ferritins are typically classified into three sub-families, DNA binding protein from starved cells (Dps), bacterioferritin (Bfr), and classical ferritin (FTn)[26]. Among them, Dps with a 12-mer core-shell structure (9 nm outer diameter) was distributed in the archaea and bacteria, whereas both Bfr and FTn groups ferritins exist as a larger 24-mer structure with a 12 nm outer diameter. The FTn as an archetypical ferritin exists in all three domains of life, while the Bfr family ferritins exist mainly in the bacteria and contain a special heme structure in its two-fold axis, which facilitates the electron transfer of the iron core[27]. In order to systematically compare ferritins with different structures from the three domains of life, eight common ferritins were selected, including Dps from bacteria (*E. coli*, EcDps) and archaea (*S. solfataricus*, ssDps), FTn from bacteria (*E. coli*, EcFTn), archaea (*P. furiosus* and *P. yayanosii*, pfFn & pyFn) and eukaryote (*H. sapiens*, heavy chain ferritin (HFn) & light chain ferritin (LFn)), Bfr from bacteria (*E. coli*, EcBfr). Firstly, multiple sequence alignments were performed to determine the sequence similarity of these ferritins. As shown in Fig. 1a, the 8 ferritins have overall low amino acid similarities. However, the phylogenetic analysis (Fig. 1b) by the neighbor-joining method revealed that most of these ferritins could be clustered into Dps, Bfr, and FTn families, which illustrated that the ferritins from the same group revealed relatively similar amino acid sequences. Yet, the *S. solfataricus* Dps was clustered with *E. coli* Bfr, which was consistent with the previous report that it owned a similar ferroxidase center to Bfr[28]. Subsequently, to evaluate the enzyme-like activities of the ferritin iron core, we first biosynthesized the ferritin core by adding the ferrous salt into the culture medium during their expression (Fig. 1c) so that the excess iron could be biomineralized in the ferritins in the form of the inorganic iron core. Then these biomineralized ferritins were purified as previously described[29–34]. As shown by the SDS-PAGE (Supplementary Fig. 1a), the biosynthetic process had a negligible effect on the ferritin monomers compared with those purified from the basic mediums. The stained transmission electron microscope (TEM) images and dynamic light scattering (DLS) spectra indicated that the biomineralized ferritins maintained their original protein-cage structures (Fig. 1d and Supplementary Fig. 1b, c). More importantly, the distinct red-brown color and the unstained TEM images of the biomineralized ferritins indicated the generation of the natural iron core in the ferritins (Fig. 1d and Supplementary Fig. 1d). However, the human light chain ferritin and the Dps sub-family members exhibit smaller core sizes and little color change, likely due to the lack of the ferroxidase center in the former and the smaller inner diameter of Dps. After their purifications, these ferritins were assessed for common nanozyme activities including SOD-, oxidase-, peroxidase-, and catalase-like activities (Supplementary Fig. 2). Interestingly, the SOD-like activity of these ferritins correlated with their origination. As shown in Fig. 1e and Supplementary Fig. 3a, both native and biosynthetic ferritins exhibited the consistent tendency that the strength of their SOD-like activities ranked in the order prokaryote > eukaryote.

To figure out the source of the SOD-like activities, thioglycolic acid and 2,2'-bipyridyl were then utilized to reduce and remove the iron core in the ferritins. As shown in Fig. 1f and Supplementary Fig. 3b–d, the SOD-like activities were largely diminished with their core removal, whereas the enzyme-like activity of the ferritins was largely unaffected after digestion of the protein shell with protease K (Fig. 1g and Supplementary Fig. 3e), proving that the iron core was responsible for the activities. Additionally, by adding gradient amounts of ferrous salt to the biosynthesis medium, it was found that the SOD-like activities of these ferritins positively correlated with their iron content (Supplementary Fig. 3f–h). Moreover, to exclude that the superoxide diminishing activity was not caused by the inhibition of the xanthine oxidase, we also detected another

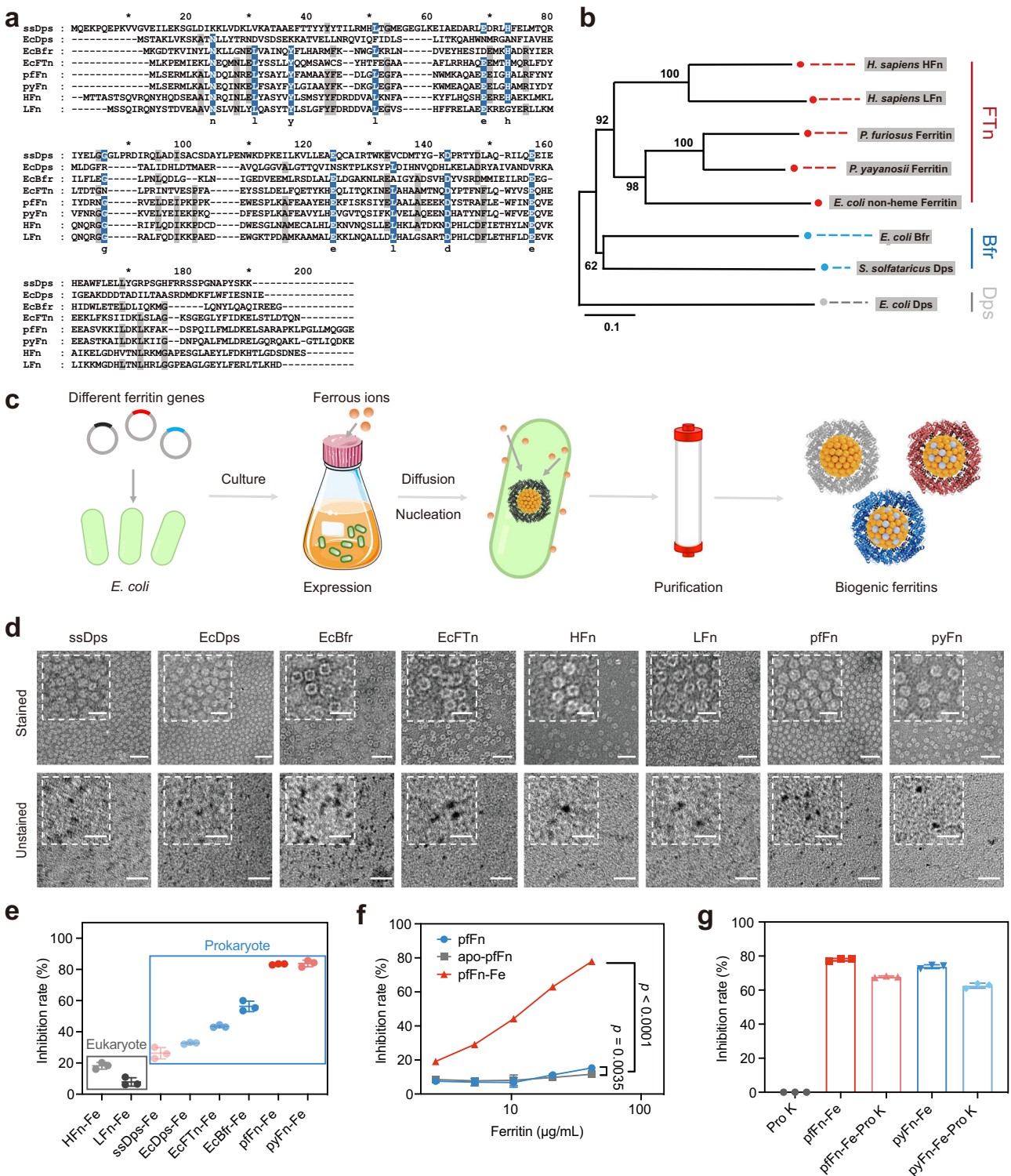

**Fig. 1 | Purification, characterization, and SOD-like activity of ferritins from different organisms. a** Multiple sequence alignment of amino acid sequences of different ferritins; the colors referred to the amino acid similarity, blue: 80%, gray: 60%. **b** Phylogenetic tree of the ferritin amino acid sequences; the numbers on the branch represented the bootstrap value. **c** Scheme of biosynthesis process of different ferritin iron cores. **d** Representative negative stained and unstained TEM images of biomineralized ferritins of three independent experiments with similar results; inset figures represented partially enlarged images; the scale bar referred to 50 nm and inset scale bar referred to 20 nm. **e** The SOD-like activity of biomineralized ferritins with the final protein concentration at 50 µg/mL; data are presented as mean ± SD ($n = 3$ independent experiments). **f** The SOD-like activity of apo-pfFn (iron-removed ferritin), pfFn, and biomineralized pfFn (pfFn-Fe); data are presented as mean ± SD ($n = 3$ independent experiments); the significant difference was evaluated by the Two-way ANOVA with post-hoc Tukey HSD test. **g** The SOD-like activity of archaeal ferritins after the protease K (ProK) digestion; data are presented as mean ± SD ($n = 3$ independent experiments). Source data are provided as a Source Data file.

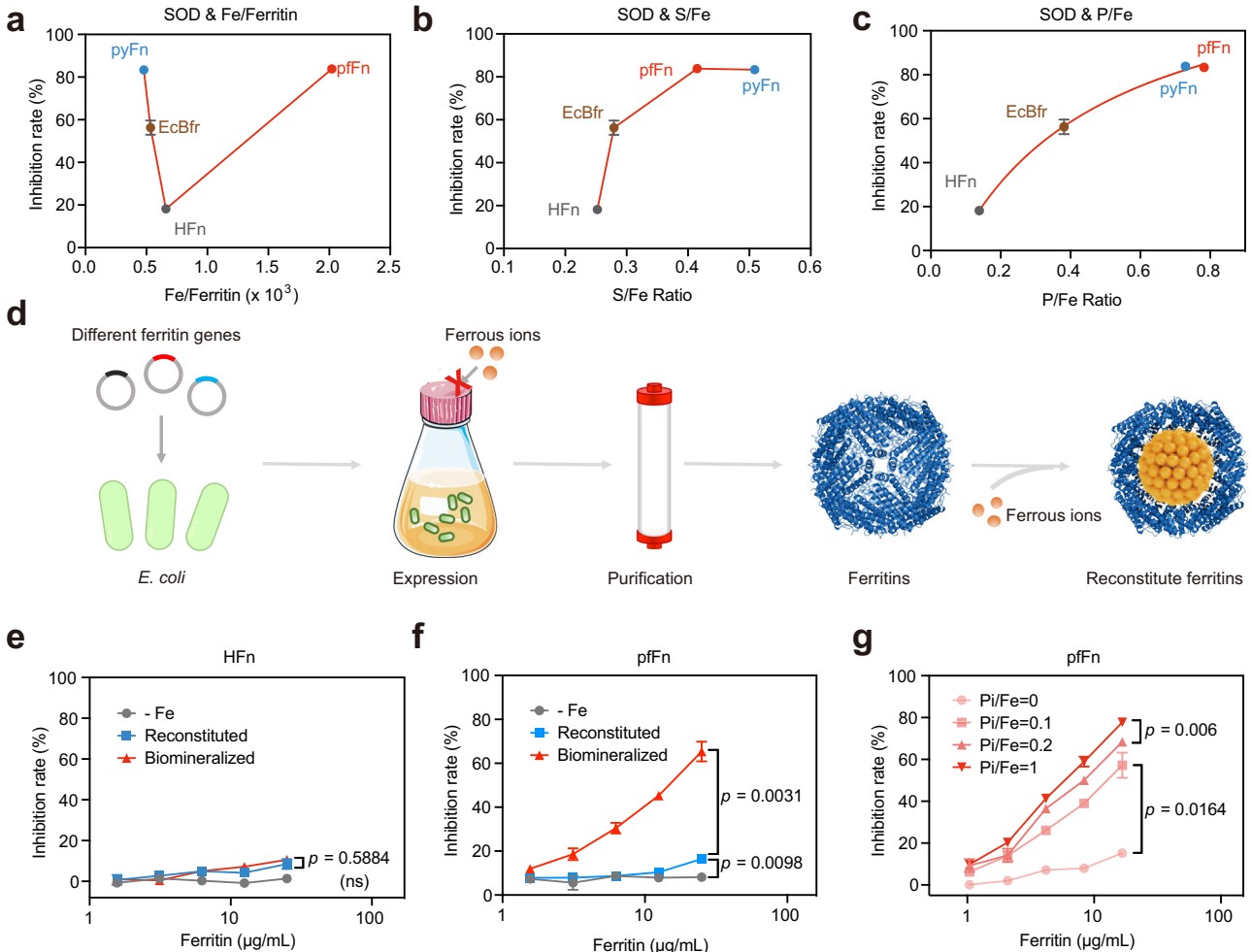

**Fig. 2 | The effect of main mineral elements in the ferritin core on the SOD-like activity.** SOD-like activity and (**a**) iron content. (**b**) S/Fe ratio and (**c**) P/Fe ratio of ferritins from different organisms; data are presented as mean ± SD (*n* = 3 independent experiments). **d** Scheme of the synthesis process of ferritin with reconstituted iron core. **e**, **f** The SOD-like activity of the reconstituted, biomineralized ferritins and ferritins purified from the basic medium (-Fe). **g** The SOD-like activity of the pfFn reconstituted with gradient phosphate/iron ratios (Pi/Fe); data in **e**–**g** are presented as mean ± SD (*n* = 3 independent experiments); the significant difference was evaluated by the Two-way ANOVA with post-hoc Tukey HSD test; ns: not significant, *p* > 0.05. Source data are provided as a Source Data file.

product-urate generation of the superoxide generation system. As shown in Supplementary Fig. 3i, j, unlike the superoxide, the urate production was not affected apparently by the biomineralized ferritins. Taken together, these results demonstrated that ferritins exhibit species-related superoxide diminishing abilities and that the iron cores are the source of the activity. Importantly, our mechanism study will prove that this activity is not simply caused by the metal ion released from ferritins.

Moreover, as the ferritins chosen here were limited to a few phyla, to further verify the relationship between the activity and sources, we then compared ferritin sequences of different phyla in relative kingdoms. As shown in Supplementary Fig. 4, the ferritin sequences in the animals of different phyla were similar. Nevertheless, there was little amino acid similarity in the ferritin sequences of bacteria and archaea ferritins in different phyla. Subsequently, we chose the other 8 ferritins from *Caenorhabditis elegans*, *Crassostrea gigas*, *Pseudomonas aeruginosa*, *Thermus scotoductus*, *Mycobacterium tuberculosis*, *Thermosphaera aggregans*, and *Thermococcus barophilus*, which involved different phyla compared with the previous 8 ferritins. Interestingly, after the purification and characterization, the SOD-like activity of these ferritins still correlated with the sources (Supplementary Fig. 5), further confirming the activity rules we found.

## The variation in SOD-like activity of ferritin iron cores resulting from differences in the iron/phosphorous ratio

As the SOD-like activity came from the iron core and positively correlated with the iron content in the same ferritins, to determine whether the discrepancies in the activity of various ferritins were due to different iron contents, inductively coupled plasma mass spectrometry (ICP-MS) was utilized to examine the major mineral elements in typical four ferritins. Unexpectedly, the iron contents of different ferritins did not correlate with their SOD-like activities (Fig. 2a), nor did the content of sulfur, an element usually present in the ferritin crystalline (Fig. 2b). Interestingly, the P/Fe ratios of iron cores were closely associated with the SOD-like activities in different ferritins (Fig. 2c), which was consistent with previous reports that ferritins from bacteria exhibited higher phosphate content than mammalian ferritins[35,36]. Next, to further verify the role of iron content, the ferritin core was reconstituted in vitro after their purification to generate ferritins with the only iron element (Fig. 2d). As except, the reconstituted HFn exhibited no significant difference with the biomineralized one that owned low P/Fe ratio (Fig. 2e), while the reconstituted archaeal ferritins exhibited markedly lower activity than the biomineralized ones (Fig. 2f), indicating that the iron was not the only factor determining the SOD-like activity. Subsequently, to verify the effect of the P/Fe ratio

on the activities, the reconstitution experiment was conducted again with ferrous salt and gradient inorganic phosphate (Pi). As shown by Supplementary Fig. 6a, the addition of phosphate apparently lightened the color of ferritins, primarily indicating that it was incorporated into the iron core. Additionally, it was found by using the ICP-MS that the incorporated iron increased initially with the addition of phosphate, presumably due to the ferrous oxidation in the ferroxidase center could be facilitated in the presence of phosphate[37], and then dropped when the phosphate concentration was high enough to form the ferric phosphate precipitation outside the ferritin (Supplementary Fig. 6b). Importantly, the ferritin reconstituted with ferrous and phosphate exhibited markedly higher SOD-like activity than those without phosphate, and the activity showed a positive correlation with the P/Fe ratio in either HFn or archaeal ferritins (Fig. 2g and Supplementary Fig. 6c–g), proving the determining role of the P/Fe ratios. These results above indicated that the difference in the SOD-like activity of ferritins results from the variance of the P/Fe ratio, and the iron core with a higher P/Fe ratio exhibits higher SOD-like activity.

### Surface potential of the 3-fold axis of ferritin affecting the iron/phosphate ratio in the core

Although it has been reported that the formation of ferritin core was affected by the environment in the cytoplasm[38], ferritins expressed in our studies were produced under the same conditions and contained different iron cores, suggesting that the structure of ferritin shapes up the iron core. To further explore the interactions, three ferritins with markedly different SOD-like activity-namely, HFn, as well as archaeal pfFn and pyFn-were reconstituted in the same buffer (molar ratio of phosphate/Fe = 1:1). Noteworthily, the activity of reconstituted HFn was apparently lower than that of pfFn and pyFn (Fig. 3a). Consistent with the alteration of activity, the P/Fe ratio of reconstituted HFn was also lower than that of pfFn and pyFn (Fig. 3b). In addition, as the HFn internalized less iron compared with pfFn and pyFn, the activities of reconstituted ferritins and biomineralized ferritins were compared in the same amount of iron to exclude the potential influence of iron variation. As shown in Fig. 3c and Supplementary Fig. 7a, their activities exhibited the same trend, demonstrating that the differences of these ferritins in SOD-like activity resulted from their discrepancy in protein structures. Interestingly, previous works about the bacterioferritin structures have found that the sulfate anion existed in the entrance of the three-fold axis of the phosphate-soaked *E.coli* bacterioferritin[39] or in the middle of the three-fold axis of the Fe-soaked *P. aeruginosa* bacterioferritin[40]. More importantly, when Fe-soaked *P. aeruginosa* bacterioferritin was soaked in the ferrous solution again, this anion sulfate would appear in the interior cavity of bacterioferritin, indicating that the 3-fold axis might be the site for the anion internalization during its iron core deposition[40,41]. Structurally, there existed hydrophobic and polar amino acids and relatively wider threefold pore in the bacterial Bfr and FTn, which might lead to the high phosphate/iron ratio in bacterial ferritin cores[42]. In support of the notion, the surface electrostatic potential of the *P. aeruginosa* Bfr 3-fold axis was less negative than that of the HFn (Fig. 3d), which apparently facilitates the anion internalization. To further understand the effect of the ferritin shell on the iron core, we mutated relative amino acids in the 3-fold axis of HFn to those of *P. aeruginosa* Bfr to increase its affinity for phosphate. Analyses with TEM, DLS spectra, SDS- and native PAGE (Fig. 3e, Supplementary Fig. 7b–d) revealed that the mutations had no effect on the whole HFn structure. However, the SOD-like activity of the mutated HFn (HFn-3-Fold-M) was higher than that of wild-type HFn (HFn-WT), and the P/Fe ratio of its iron core increased markedly (Fig. 3f, g and Supplementary Fig. 7e). In the reconstitution experiments, the HFn mutant also exhibited higher SOD-like activity and phosphate intake ability than the wild-type HFn (Supplementary Fig. 7f), which illustrated the effect of the protein structure on the core activities and compositions.

In addition, as mammalian ferritins usually exist as heteropolymers, to further verify the correlation of the ferritin structures and SOD-like activities, the natural homopolymers ferritin in human-mitochondrial ferritin (MFn) and H/L heteropolymers ferritins (H/LFn) were biomineralized and purified (Supplementary Fig. 7g, h). Likewise, the biomineralization process did not affect the structure and there was obvious core formation after the biosynthesis process (Fig. 3h and Supplementary Fig. 7i–k). More importantly, the MFn and H/LFn also exhibited low SOD-like activity as HFn either in the biomineralized (Fig. 3i) or the reconstituted experiments (Fig. 3j and Supplementary Fig. 7l). To illustrate the specific activity discrepancy, we also calculate the SOD units based on commercial human CuZnSOD or the method offered by the SOD assay kit. As seen in Supplementary Fig. 7m, n, the SOD activities of reconstituted human ferritin were near 1000 U/mg Fe, however, the archaea ferritins could arrive at 2000−3000 U/mg.

From the above results, it is reasoned that protein structure, especially the three-fold axis potential, played a key role in the catalytic activity of the ferritin iron core. To further investigate whether the potential discrepancy of HFn and *Pseudomonas aeruginosa* Bfr was universal, we searched the PDB database and calculated 20 bacteria ferritins and 20 eukaryote ferritins. Remarkably, all 20 eukaryote ferritins revealed more negative potential in the three-fold axis than those of all the 20 bacteria ferritins (Supplementary Figs. 8 and 9), disclosing the structure rule between the bacteria and eukaryote ferritins.

### Different iron/phosphate ratio leading to the structure change of the iron core

As phosphate significantly affected the activity of the iron core, we thus explored the difference of the iron core with or without phosphate. Interestingly, it has been found that the iron core of human spleen ferritin formed a more ordered structure than those from limpet and *P. aeruginosa*[43], which prompted us to examine the structure of the ferritin iron cores with different SOD-like activity. Under high-resolution transmission electron microscopy (HRTEM), the iron core in HFn exhibited mostly as single-domain crystallite, whereas, it apparently switched to the polycrystalline structure when phosphate was reconstituted into the core. The iron core of archaeal ferritin that possesses a higher P/Fe ratio appeared as the completely amorphous domain (Fig. 4a and Supplementary Fig. 10a). These data indicated that the incorporation of phosphate into the iron core changes its structure and at the same time, augments its catalytic activity.

Iron deposited in ferritin was reported to be in the form of ferric oxyhydroxide or ferrihydrite[44–46]. To further confirm the effect of the phosphate on the morphology and activity of the iron core, phosphate-doped 2-line ferrihydrite was first synthesized to mimic the ferritin iron core containing various amounts of phosphate. As shown in the TEM images and DLS spectra, the ferrihydrite with or without phosphate both displayed the ~200 nm amorphous structure (Supplementary Figs. 10b and 11a). The marked color change and the appearance of phosphate peak at ~1050 cm$^{-1}$ in the Fourier transform infrared spectroscopy (FTIR) spectrum demonstrated the successful incorporation of the phosphate (Fig. 4b and Supplementary Fig. 10b). X-ray photoelectron spectroscopy (XPS) was also deployed to detect the chemical state and the overall electronic structure of elements. As shown in Supplementary Fig. 11b–g, the valence of iron and phosphorus remained $Fe^{3+}$ and $P^{5+}$ respectively in all the ferrihydrite materials. X-ray diffraction analysis (XRD) was performed to characterize the crystallographic structure of these ferrihydrites. As shown in Fig. 4c, with the addition of the phosphate, the second peak in the XRD spectra diminished gradually, primarily suggesting the loss of crystallinity. The same tendency was found by analyses with HRTEM and SAED. The doped phosphate caused the lattice structure to transit from the single domain crystallite to amorphous and the disappearance of the diffraction spots (Fig. 4d), which were highly consistent with results in ferritins. Taken together, these data indicated

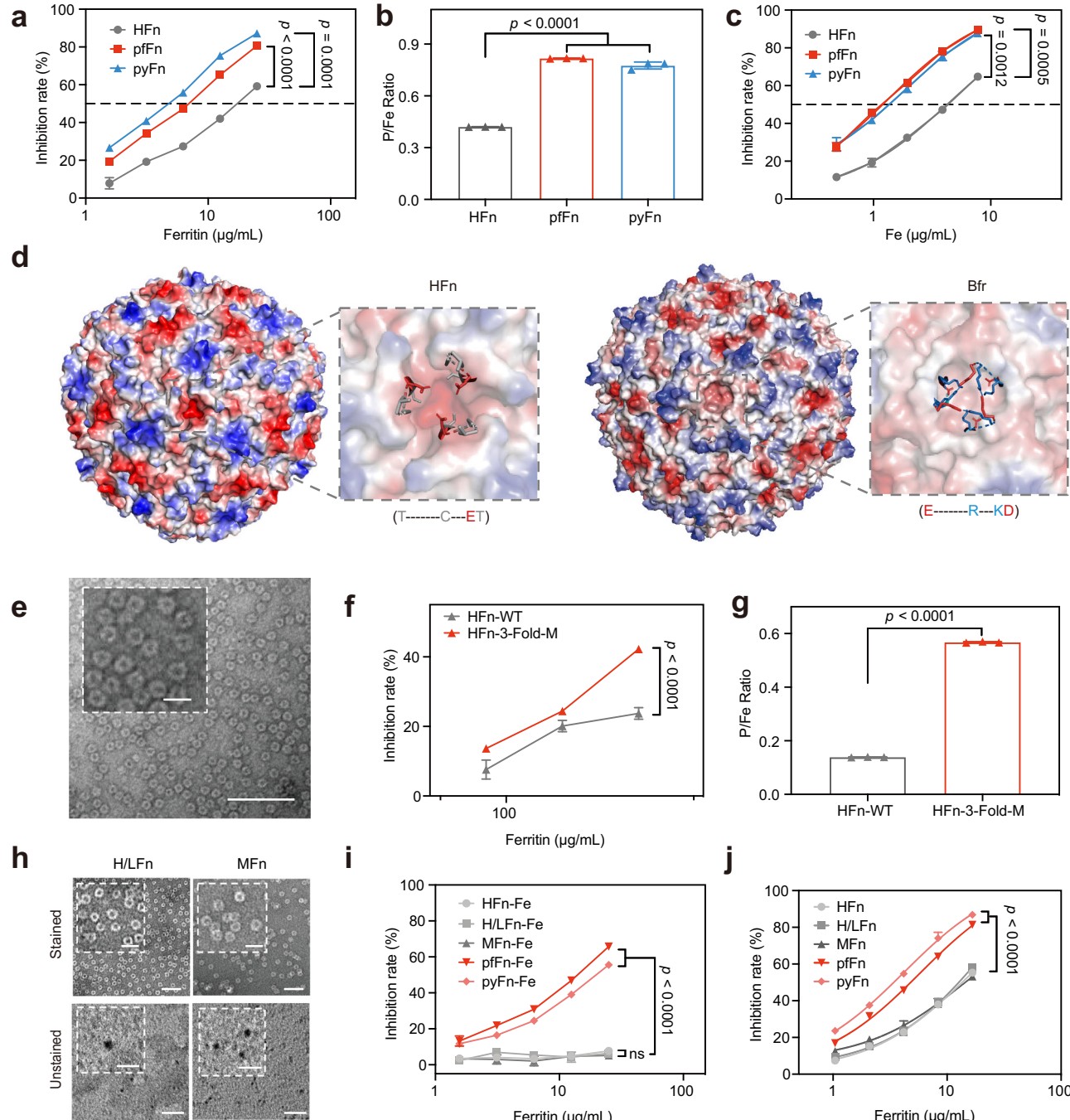

**Fig. 3 | Effect of protein structure on SOD-like activity and iron/phosphorous ratio of ferritin. a** SOD-like activity and (**b**) P/Fe ratio of HFn, pfFn and pyFn reconstituted in the same buffer (molar ratio of Pi/Fe = 1:1). **c** SOD-like activities of reconstituted HFn, pfFn and pyFn in the same total iron content. **d** The surface potential of the whole structure and three-fold axis of the HFn and Bfr (*P. aeruginosa*) with the amino acids around the three-fold axis marked in the bottom; the colors referred to electrostatic potential, red: negative, white: neutral, blue: positive. **e** Representative negative stained TEM image of the three-fold pore mutated HFn of three independent experiments with similar results; inset figures represented partially enlarged images; the scale bar referred to 50 nm, inset scale bar referred to 20 nm. **f** SOD-like activity and (**g**) P/Fe ratio of the biomineralized WT (HFn-WT) and mutated HFn (HFn-3-fold-M). **h** Negative stained and unstained TEM images of the H/LFn and MFn of three independent experiments with similar results; inset figures represented partially enlarged images; the scale bar referred to 50 nm, inset scale bar referred to 20 nm. **i** SOD-like activity of biomineralized human HFn, MFn, H/LFn and the archaeal pfFn, pyFn. *p* valve between HFn-Fe and H/LFn-Fe was 0.8465, and between HFn-Fe and MFn-Fe was 0.1187. **j** SOD-like activity of reconstituted human HFn, MFn, H/LFn, and the archaeal pfFn, pyFn. All the data are presented as mean ± SD (*n* = 3 independent experiments). The significant differences were evaluated by the Two-way ANOVA with post-hoc Tukey HSD test for (**a, c, i, j**), by the One-way ANOVA with post-hoc Tukey HSD test for (**b**), and by Two-sided unpaired Student's *t*-test for (**g**); ns: not significant, $p > 0.05$. Source data are provided as a Source Data file.

that the phosphate incorporated in the ferritin changes the structure of the iron core in a concentration-dependent manner.

The SOD-like activity of these ferrihydrites was then examined in either the same quality or the same iron content. As shown in Fig. 4e

and Supplementary Fig. 11h, the activity positively correlated with the P/Fe ratio in these materials. Moreover, as the ferrihydrite in the ferritins exists transformation from 2-line to 6-line[47], the phosphate-doped 6-line ferrihydrite was also synthesized (Supplementary

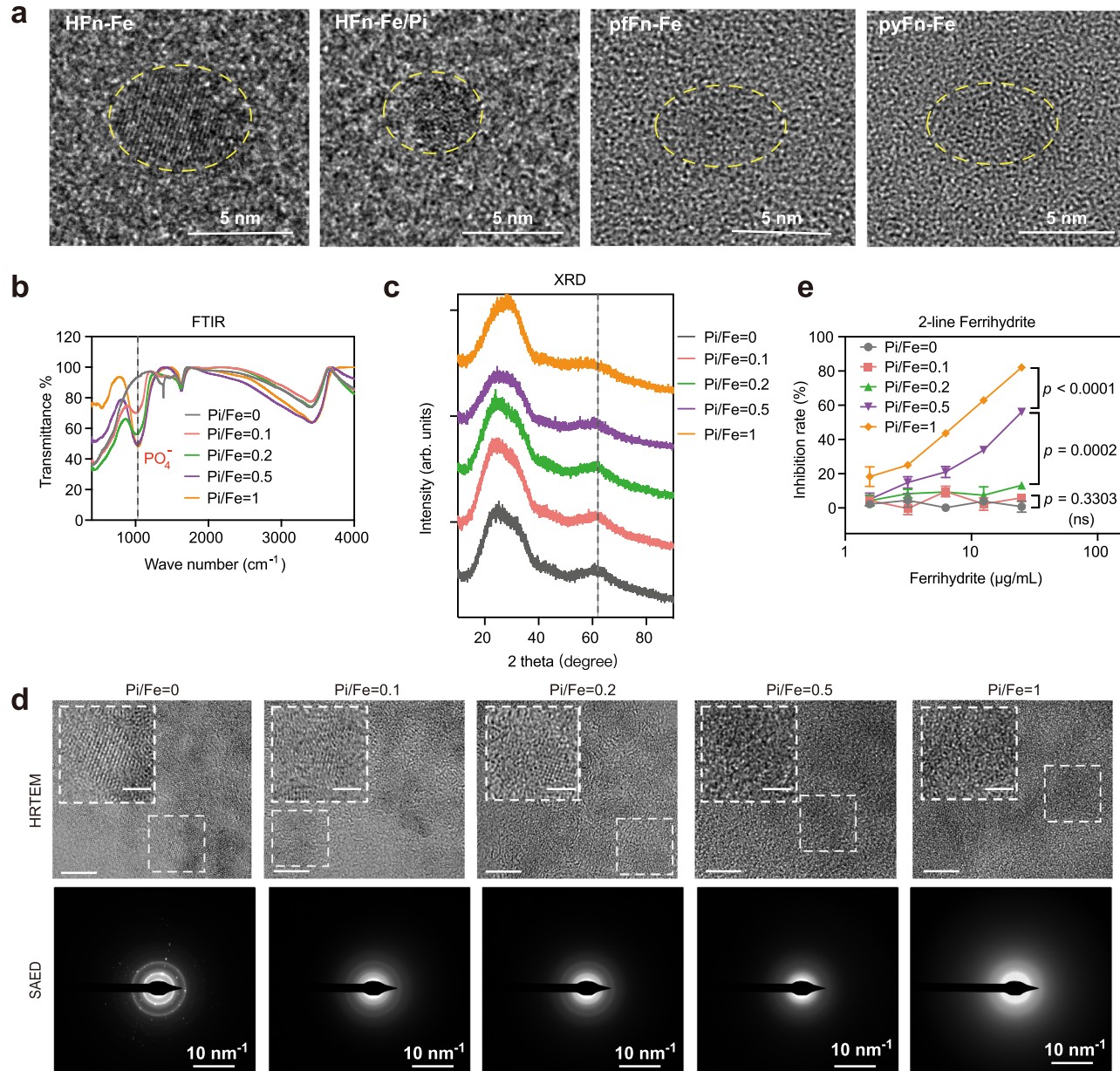

**Fig. 4 | Characterization of the ferritin iron core and phosphate-doped ferrihydrite. a** HRTEM images of ferritin iron core with different P/Fe ratios of three independent experiments with similar results. **b** FTIR spectra of 2-line ferrihydrite doped with gradient phosphate. **c** XRD spectra of 2-line ferrihydrite doped with gradient phosphate. **d** HRTEM and SAED images of 2-line ferrihydrite doped with gradient phosphate of three independent experiments with similar results; inset figures represented partially enlarged images; the scale bar in HRTEM images referred to 5 nm; the inset scale bar referred to 2 nm. **e** SOD-like activity of the phosphate-doped 2-line ferrihydrite; the data are mean ± SD ($n = 3$ independent experiments); the significant difference was evaluated by the Two-way ANOVA with post-hoc Tukey HSD test; ns: not significant, $p > 0.05$. Source data are provided as a Source Data file.

Fig. 12a). Similar to the 2-line ferrihydrite characterization, the FTIR and XRD analyses illustrated the successful incorporation of the phosphate (Supplementary Fig. 12b, c). More importantly, the SOD-like activity also increased with the addition of the phosphate (Supplementary Fig. 12d), further demonstrating the important role of the phosphate on the ferrihydrite structure and activity.

## Phosphate coordinated with the iron atom leading to high SOD-like activity

It has been previously shown that phosphate exists on either the surface of the iron core in horse spleen ferritin or in the interior of the bacteria ferritin[36], which suggests the different locations of the phosphate in the ferritin core. To investigate the specific effect of the surface phosphate, the iron-reconstituted ferritins and the 2-line

ferrihydrite were exposed to surface-binding phosphate. Unlike those reconstituted with phosphate, the color of the surface binding ferritins did not change (Supplementary Fig. 13a), implying that the surface phosphate has not altered the iron core structure. In addition, the ferritin prepared from the gradient phosphate buffer did not exhibit increased SOD-like activity (Fig. 5a and Supplementary Fig. 13b–d). Additionally, ferrihydrites absorbed gradient phosphate also showed negligible changes in their SOD-like activities, even though the P/Fe ratio has increased close to 1 (Fig. 5b, c). These results indicated that surface-absorbed phosphate does not contribute to the SOD-like activity of the ferritin iron core.

To further define the location of phosphate in the iron core, the Fe K-edge X-ray absorption fine structure (XAFS) spectroscopy was used to analyze the chemical state and coordination environment of the

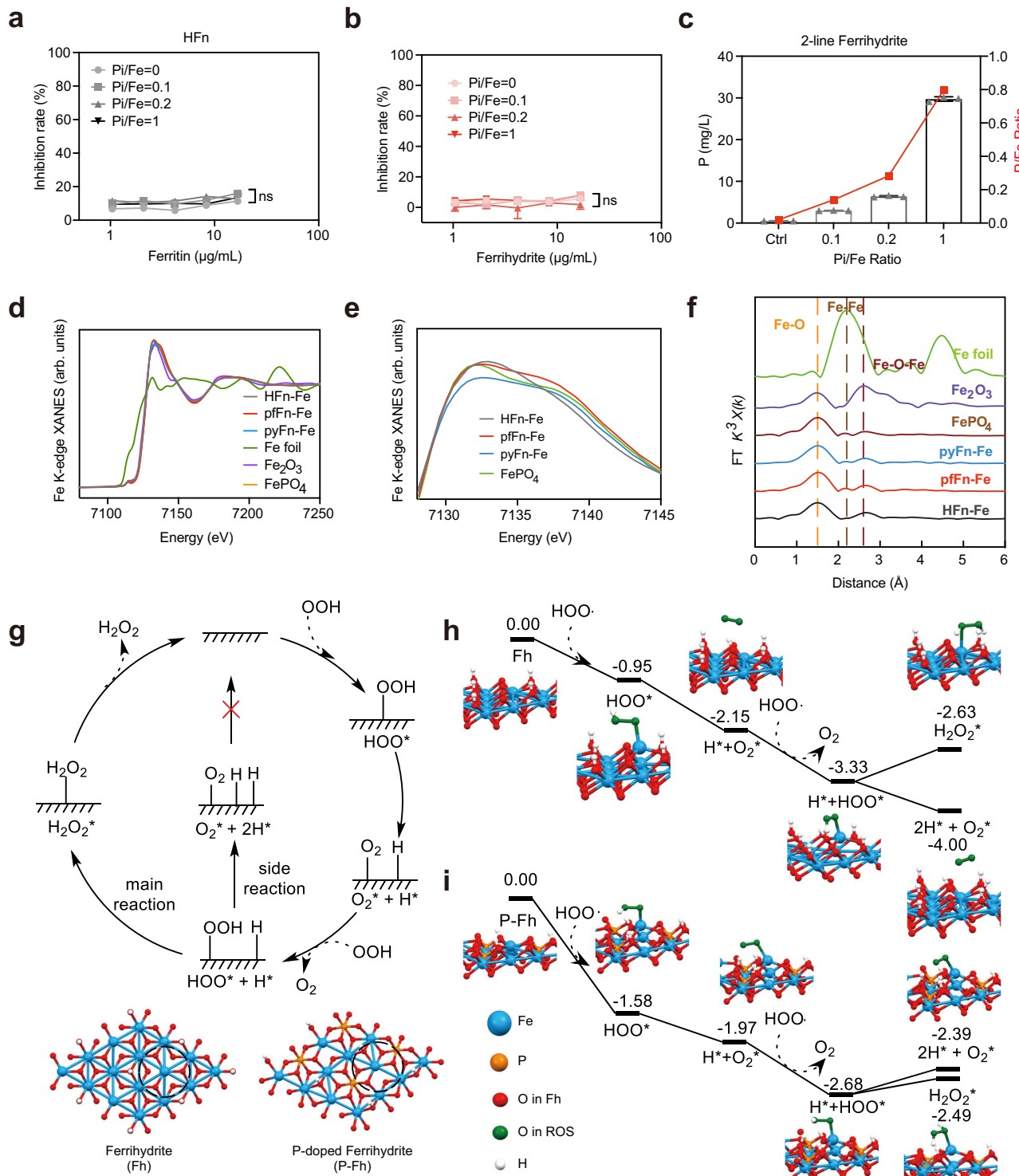

**Fig. 5 | Atomic structure and DFT analysis of ferritin iron cores. a** SOD-like activity of reconstituted HFn with gradient surface binding phosphate. The $p$ value of Pi/Fe = 0.1, 0.2, 1 compared with Pi/Fe = 0 was 0.091, 0.435, 0.5339, respectively. **b** SOD-like activity of the 2-line ferrihydrite with gradient surface binding phosphate. The $p$ value of Pi/Fe = 0.1, 0.2, 1 compared with Pi/Fe = 0 was 0.7172, 0.7761, 0.4216, respectively. The data in (**a**, **b**) are mean ± SD ($n$ = 3 independent experiments); the significant difference was evaluated by the Two-way ANOVA with post-hoc Tukey HSD test; ns: non-significance, $p$ > 0.05. **c** ICP-MS of P content and P/Fe ratios in the 2-line ferrihydrite with gradient surface binding phosphate; the data are presented as mean ± SD ($n$ = 3 independent experiments). **d**, **e** XANES spectra of

biomineralized HFn, pfFn, pyFn and relative reference materials. **f** Fourier-transformed magnitudes of the experimental Fe K-edge EXAFS signals of biomineralized HFn, pfFn, pyFn, and relative reference materials. **g** Proposed two possible reaction pathways for superoxide with products of $H_2O_2$ and $O_2$ (main reaction) or $O_2$ and $H^*$ (side reaction) and catalytic centers in Ferrihydrite and P-doped Ferrihydrite. **h** Reaction profiles with key intermediate structures and reaction energies (eV) for ferrihydrite. **i** Reaction profiles with key intermediate structures and reaction energies (eV) for P-doped ferrihydrite. Only important structural fragments were shown for clarity. Source data are provided as a Source Data file.

biomineralized HFn, pfFn and pyFn. As shown by the X-ray absorption near edge structure (XANES) profiles, the absorption edges of all the ferritins were similar to the $Fe_2O_3$ and $FePO_4$, indicating that the Fe in the ferritins exhibited a similar electronic structure as these two referred materials (Fig. 5d). On the other hand, the pre-edge peak of all the ferritins suggested the octahedral structure of the iron atom[48]. Furthermore, compared with that of HFn, there was a shoulder at ~7137 eV in the XANES profile of archaeal ferritins that was also presented in the $FePO_4$ standard (Fig. 5e), primarily suggesting the presence of phosphorus in the surrounding environment of the iron atom. The extended region of the XAFS spectra was used to gain the specific structure information on the atomic shell around the iron[49]. As shown in Fig. 5f, all the ferritins exhibited a major peak at 1.5 Å that is attributed to the Fe-O coordination in the first core shell, which agrees nicely with the presence of $Fe_2O_3$ and $FePO_4$. Moreover, all three ferritins had the same $Fe_2O_3$ peak at 2.6 Å which is attributable to the presence of the Fe-O-Fe coordination in the second shell. However, compared with HFn, there was a minor peak at 2.2 Å in the spectrums of pfFn and pyFn, which agreed with the $FePO_4$ structure and suggested that some phosphorus had replaced the iron atom in the second shell. Quantitative EXAFS curve-fitting analysis was then performed to investigate the coordination configuration (Supplementary Fig. 14a and Supplementary Table 1). The proper fit model for all the ferritins was two coordination shells in which the first shell with 2 Å diameter was occupied by six oxygen atoms and nearly 2 iron atoms accounted for the second shell at a 3 Å distance. In addition, there existed the Fe-P coordination shell at 2.6 Å diameter for the archaeal ferritins but not the HFn, indicating that some phosphorus replaced the iron in the second shell. Taken together, these results illustrated that the major difference of biosynthetic HFn core and archaeal ferritin pfFn and pyFn core is the phosphorus coordination in the second shell of the iron atom, which likely affects the reaction ability with superoxide.

In order to figure out how the coordinated phosphate promoted the SOD-like activity, we carried out DFT calculations to uncover the reaction processes on ferrihydrite and phosphate-doped ferrihydrite. Because superoxide anion $O_2^{\cdot-}$ readily protonates to form $HOO^{\cdot}$ in aqueous condition[50], the $HOO^{\cdot}$ radical was used to represent $O_2^{\cdot-}$ in the DFT calculations. As shown in Fig. 5g, the proposed pathway involves the adsorption of $HOO^{\cdot}$ and its subsequent disproportionation to form $O_2$ and $H_2O_2$. Our previous study[51] has shown that these reactions can occur with low barriers. Thus, we considered only the adsorption of important intermediates on the catalyst surface. A slab model of the ferrihydrite (001) surface was adopted in these calculations, where the active site was circled with a dashed line (Fig. 5g and Supplementary Fig. 14b). Figures 5h, i showed the corresponding energy profiles for the disproportionation of $HOO^{\cdot}$ on ferrihydrite and P-doped ferrihydrite, respectively. It is worth noting that the adsorption of the first $HOO^{\cdot}$ and subsequent O-H splitting to release $O_2$ are exothermic, indicating that they can easily occur on both ferrihydrite and P-doped ferrihydrite. However, the formation of $H_2O_2$ after adsorption of the second $HOO^{\cdot}$ on ferrihydrite is endothermic with an increase in energy by 0.7 eV (Fig. 5h). In contrast, the generation of $O_2$ is exothermic with energy decreased by 0.67 eV (Fig. 5i). This is probably due to the (001) surface of ferrihydrite having a very strong affinity to H, which made the H-adsorbed structure very stable. The result of Fig. 5h suggested that ferrihydrite tended to oxidize $O_2^{\cdot-}$ to form $O_2$ rather than catalyzed its disproportionation of $O_2^{\cdot-}$ like a SOD. On the other hand, the formation of $H_2O_2$ was energetically more favorable than that of $O_2$ by 0.29 eV in P-doped ferrihydrite (Fig. 5i), indicating that the phosphate-doped ferrihydrite can readily catalyze the disproportionation of superoxide. These DFT results agree with the experiment that ferrihydrite has a superior SOD-like catalytic activity by phosphate doping.

## The catalytic feature of the SOD-like activity of ferritins

In order to verify the catalytic role of the ferritin iron core, the reaction processes with superoxide were subsequently examined. Firstly, the intermediate product ferrous ion was tracked by the specific chelator ferrozine. As shown in Fig. 6a, the ferrous ion was detected during the reaction of ferritins with the xanthine oxidase (XOD)/xanthine (Xan) and exhibited a positive correlation with the xanthine levels (Fig. 6b). Moreover, there was no ferrous ion generated without the XOD or xanthine, indicating the ferrous ion mainly came from the reduction by the superoxide (Supplementary Fig. 15a–c). However, the ferrous ion became undetectable when ferrozine was added after the reaction, likely due to the re-oxidation of the ferrous ions (Fig. 6c). These results were also supported by previous reports that the iron core remains mostly inside ferritin after ferric reduction and could be re-oxidized subsequently[35,52,53]. The same results were obtained in experiments with phosphate-doped ferrihydrite, in which the doped phosphate enhanced the reaction ability of ferrihydrite with the superoxide, and there was also no ferrous ion detected after the reaction (Supplementary Fig. 15d, e). To distinguish the SOD-like activity of ferritin with the ferric ion catalyzed Haber-Weiss reaction, we estimated based on the standard ferrous concentration curve that the generated ferrous ions accounted for 0.44%, 3.07%, 1.23% of the iron in the HFn, pfFn and pyFn respectively in the time course of SOD activity test (Fig. 6d). In contrast, the superoxide diminishing ability of these ferric ions was much lower than their corresponding ferritins (Fig. 6e). Furthermore, superoxide diminishing efficiency of ferric salt in the equivalent total iron level was also investigated. As shown in Supplementary Fig. 15f, the SOD activity of the ferric salt in the same total iron level was much lower than the archaea but higher than the HFn group, suggesting that the SOD-like activity dose not result from the ionic reaction. Additionally, the generation of ferrous ions from archaeal ferritins and ferric salt in the same total iron content was also evaluated. As shown in Supplementary Fig. 15g, compared with the ferritins, the ferric salt produced much more ferrous while on the contrary exhibited low superoxide diminishing ability. Furthermore, unlike the reaction with ferritin, there remained a high level of ferrous after the reaction (Fig. 6f), suggesting that the ferritins deplete the superoxide in a way different from the ferric. Electron spin resonance (ESR) was then utilized to evaluate the ability of ferritins to deplete superoxide. As expected, the ferritins significantly decreased the amount of superoxide in a dose-dependent manner (Fig. 6g and Supplementary Fig. 15h, i), and the archaea ferritins exhibited a stronger ability than HFn in reducing superoxide (Fig. 6h). Of note, the ferritin-catalyzed superoxide decay was completely different from the ferric ion-induced reaction, as the latter produced obvious hydroxyl radicals (Supplementary Fig. 15j). We further assessed the SOD-like activity of pyFn-Fe and phosphate-doped ferrihydrite after their reactions with gradient amounts of xanthine. Both of them exhibited consistent SOD-like activity (Fig. 6i), and their iron contents did not decrease with the reaction (Supplementary Fig. 15k, l), further validating the catalytic role of the iron core. In summary, these results demonstrated the stronger superoxide-depleting ability of archaeal ferritin when compared with that of HFn and showed that the ferritin iron core catalyzes the superoxide depletion in a way different from ferric ions.

## Antioxidant ability of the ferritin in transgenic and mutated *E. coli*

Inspired by the SOD-like activity of the prokaryote ferritins, we investigated their antioxidant ability in transgenic *E. coli* exposed to paraquat, which could stimulate superoxide production[54]. As shown in Supplementary Fig. 16a, BL21(DE3) competent cells transformed with plasmids encoding HFn, pfFn, pyFn expressed similar levels of the ferritins. Next, to detect the superoxide sensitivity of these strains, the spot assay was conducted by dropping serially diluted transgenic

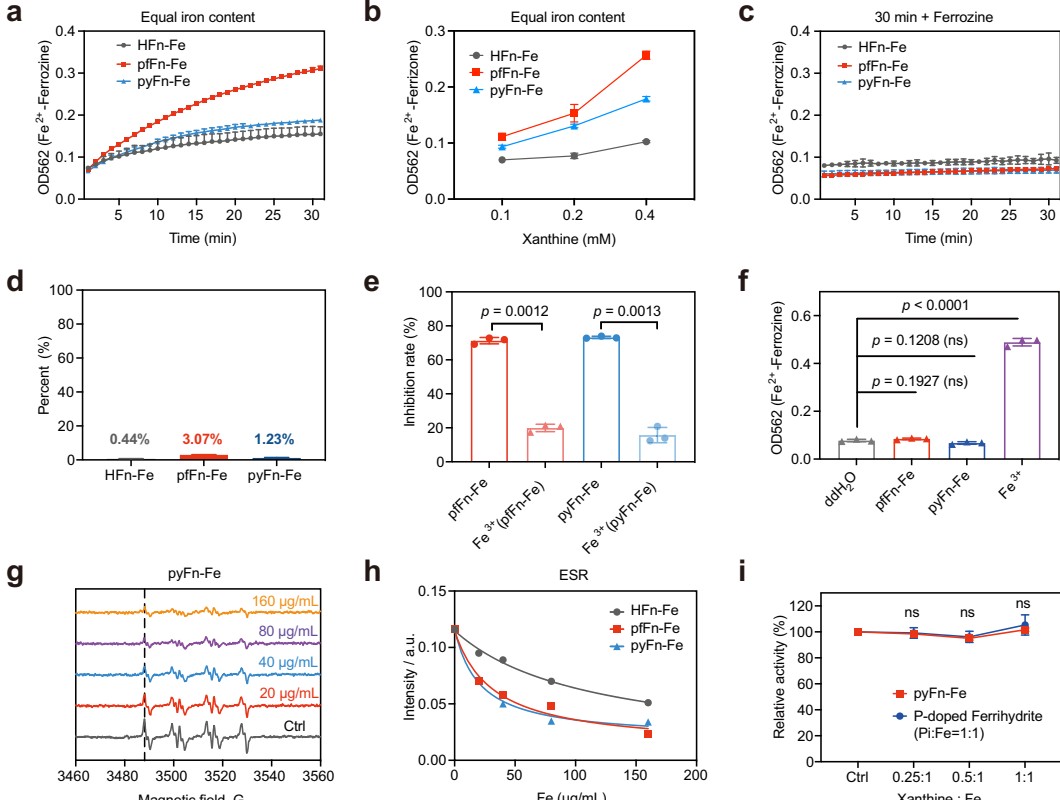

**Fig. 6 | Characterization of the reaction process of the SOD-like activity of ferritins. a** Ferrous-ferrozine production curve in the reaction of ferritins with XOD-Xan. **b** Content of the ferrous-ferrozine production in the reaction of ferritin with XOD and gradient xanthine. **c** The ferrous-ferrozine production in the reaction of ferritin and XOD-Xan when the ferrozine was added 30 min later than xanthine oxidase. **d** The proportion of intermediate ferrous ions to the whole iron content in different ferritins after their reaction with XOD-Xan. **e** SOD-like activity of the archaeal ferritins and relative ferric ion, which was generated in the reaction of archaeal ferritins with XOD-Xan in the 30 min course. **f** The ferrous production of archaeal ferritin and ferric salt after the reaction with the XOD-Xan in the iron concentration at 44.8 μg/mL. **g** ESR spectra of the DMPO/HOO• in the presence of XOD/Xan and gradient pyFn; the marked concentration referred to iron content. **h** The ESR intensity of DMPO/HOO• of HFn, pfFn and pyFn groups. The line at 3488.125 G of DMPO/ HOO• spectra was used to represent the intensity. **i** SOD-like activity of the pyFn and phosphate-doped ferrihydrite after their reaction with gradient xanthine. All data in (**a**–**f, i**) are mean ± SD ($n = 3$ independent experiments). The significant difference was evaluated by Two-tailed unpaired Student's $t$-test for (**e**), and by One-way ANOVA with post-hoc Tukey HSD test for (**i**). The $p$ value in (**i**) compared with Ctrl for pyFn group from left to right were 0.7532, 0.0742, 0.1398; and for phosphate-doped ferrihydrite were 0.9967, 0.7771, 0.5676; ns: not significant, $p > 0.05$. Source data are provided as a Source Data file.

bacteria onto the plate containing paraquat. As shown in Fig. 7a, transgenic bacteria expressing the archaeal ferritins exhibited markedly higher resistance to paraquat than those transformed with the empty vector or the HFn-encoding vector. While all the bacteria grew at a similar rate without paraquat, bacteria transformed with plasmids encoding pyFn or pfFn grew significantly faster than those transformed with empty vector or plasmid encoding HFn in the presence of paraquat (Fig. 7b, c), illustrating higher superoxide tolerance of the pyFn or pfFn groups. The bacteria viability was assessed by the colony formation experiment. As shown in Fig. 7d, there remained more single colonies of the pyFn or pfFn groups compared with the empty vector and HFn group. Likewise, the viabilities of bacteria were also reflected by the dehydrogenase activity of bacteria, which could reduce the WST-8 indicator and exhibit a specific absorption peak at 450 nm. As shown in Supplementary Fig. 16b, the bacteria transformed with plasmids expressing pfFn and pyFn exhibited markedly higher dehydrogenase activities in the presence of paraquat. A similar result was obtained by the confocal laser scanning microscopy (CLSM) image after co-staining with SYTO-9 (living bacteria) and propidium (PI, dead bacteria) (Supplementary Fig. 16c). The morphology of bacteria was further analyzed by scanning electron microscope (SEM). The bacteria in all the different groups exhibited the typical rod-like shape with intact and smooth cell walls. While the bacteria in the control groups suffered significant membrane damage and cell shrink after paraquat

treatment, those expressing HFn had relieved paraquat damage but still showed moderate shrinkage. In contrast, the expression of pfFn and pyFn largely reversed the effects of paraquat (Fig. 7e). Additionally, replating the paraquat-treated bacteria on the LB plates also verified the high cell viabilities in the pfFn and pyFn transformed groups. Next, we used the fluorescent probe 2'-7'-dichlorofluorescein diacetate (DCFH-DA) to detect ROS levels in paraquat-exposed bacteria. As expected, the bacteria containing pfFn and pyFn showed significantly lower levels of ROS in comparison with those transformed with the empty vector or vector expressing HFn (Fig. 7g). These results demonstrated that consistent with their SOD-like activity, the prokaryote ferritins exert strong antioxidant function in bacteria and may have important physiological functions.

Subsequently, to further verify the different superoxide-diminishing abilities of ferritins in human and bacteria, we both substituted the *Bfr* gene with the *HFn* gene or knocked out the *Bfr* gene in the BL21(DE3) strain by the CRISPR/Cas9 system[55] (Supplementary Fig. 16d). As shown in Supplementary Fig. 16e, after the donor fragments with the *HFn* gene and the plasmids containing the gRNA and Cas9 were transformed into the BL21 strain, colonies with double resistance were selected. Next, by using the genomic forward primer and reverse primer inner the *HFn*, the colonies with successful gene editing were selected for the plasmids curing and the electrophoresis result indicated the obvious gene substitution. Likewise, the knockout

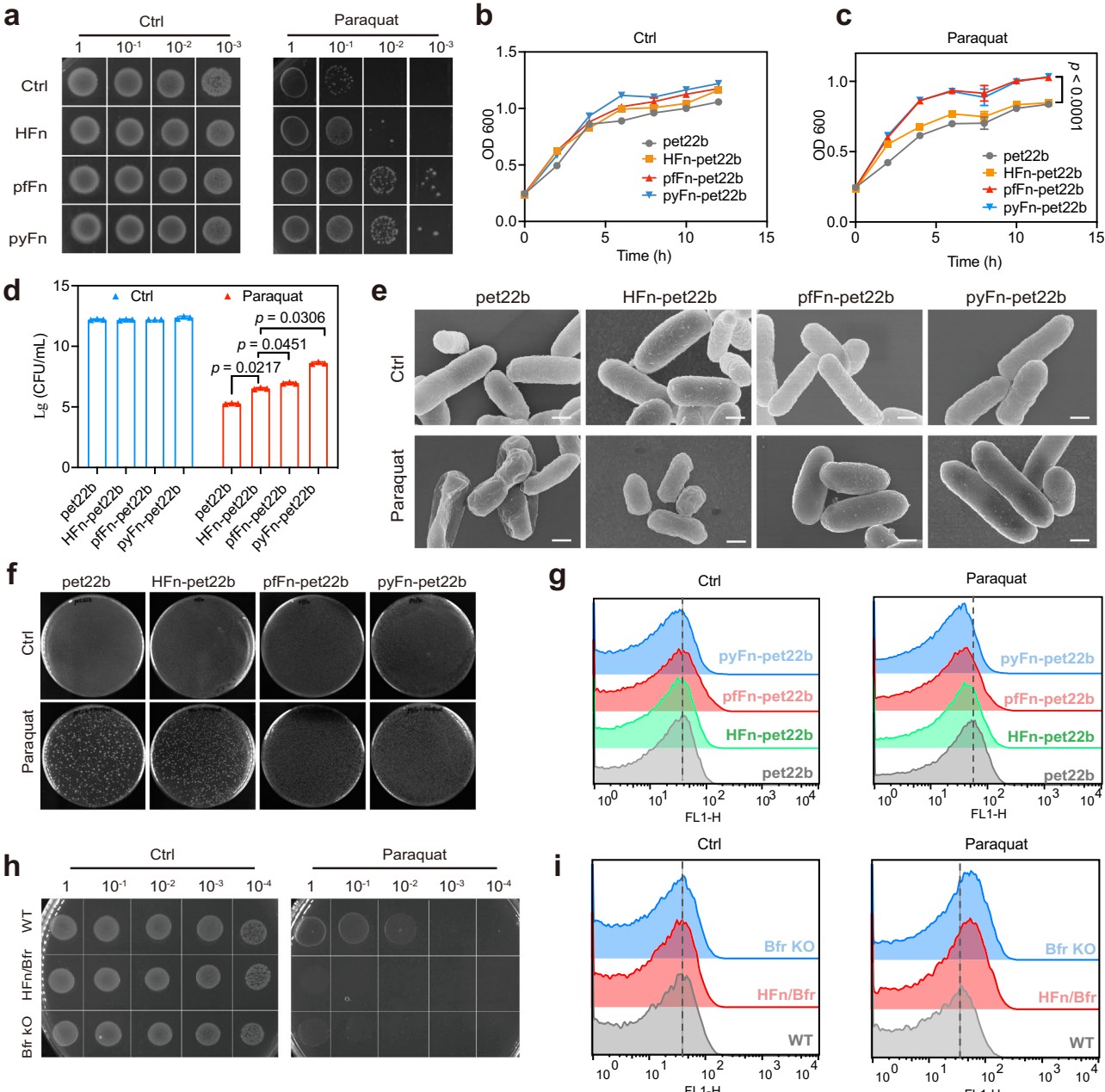

**Fig. 7 | The antioxidant ability of ferritin in transgenic *E. coli*. a** Spot assay of the different transgenic *E. coli* under the pressure of paraquat of three independent experiments with similar results. Ctrl referred to groups without paraquat treatment. **b, c** The growth curve of different ferritin transgenic *E. coli* with or without the paraquat stimulation; the data are mean ± SD (*n* = 3 independent experiments); the significant difference was evaluated by the Two-way ANOVA with post-hoc Tukey HSD test. **d** The colony formation amount of the ferritin transgenic *E. coli* with or without the paraquat stimulation; the data are mean ± SD (*n* = 3 independent experiments); the significant difference was evaluated by the One-way ANOVA with post-hoc Tukey HSD test. **e** Representative SEM images of ferritin transgenic *E. coli* after the paraquat treatment of three independent experiments with similar results; the scale bar referred to 0.5 μm. **f** Digital images of replated ferritin transgenic *E. coli* after the paraquat treatment of three independent experiments with similar results. **g** ROS levels of the different ferritin transgenic *E. coli* after the paraquat treatment. **h** Spot assay of the WT, *Bfr* knockout (*Bfr* KO) and Bfr substitution strain (*HFn/Bfr*) of three independent experiments with similar results. **i** ROS levels of WT, *Bfr* knockout and substitution strain after the paraquat treatment. Source data are provided as a Source Data file.

strain was acquired by the homologous fragment without *HFn* and the knockout efficiency was verified by the upstream and downstream primers in the genome of the *Bfr* gene (Supplementary Fig. 16f). After the successful construction of these two strains, their antioxidant abilities against paraquat were compared. As shown by the spot assay (Fig. 7h), the *Bfr* knockout apparently increased the superoxide sensitivity of the BL21 strain and the *HFn* substitution did not contribute to the antioxidant abilities. Moreover, by detecting the ROS levels, we also found higher ROS levels in both the *Bfr* knockout and substitution strains, further indicating the antioxidant function of the ferritins with SOD-like activity (Fig. 7i).

## Discussion

The biominerals are widely distributed and play an important physiological role in different organisms. However, they mainly exploited physical-related properties like mechanical support and optical or magnetic sensing in vivo. The rise of nanozymes implied that nano-biominerals might function as enzyme-like catalyzers. In this study, we

chose ferritins as the research model. By comparing the SOD-like activities of ferritins from three living kingdoms, we found for the first time the relationship between the superoxide scavenging ability of ferritin iron cores and their sources of species, which is in the order of prokaryote > eukaryote. It is conceivable that the difference reflects the imminent requirement for cells of unicellular organisms to survive the oxidative environment, to which most of the cells in higher organisms are not exposed directly.

By identifying the core constitution, we found that the difference in the activity was caused by the phosphate coordinated with the iron atom inner the ferritin, which was initially shaped by the structure. In fact, previous studies have found higher inorganic phosphate content existed in the ferritins of bacteria and limpet hemolymph than those of human spleen[43] and *A. vinelandii* ferritin had phosphate neighbored to the iron atom in the core, compared with horse spleen ferritins, which agreed well with our study[36,44]. Of note, it was speculated in previous literature that this variance in the bacteria and mammalian ferritin core was caused by the different phosphate content in the cytoplasm. Our reconstitution and mutant experiments indicated that the protein structure was also the decisive factor for the variance of the iron core.

It is interesting to know how phosphate in the core affects enzyme-like activity. It was reported that the phosphate in the ferritin core affected the oxidant and deposition of the ferrous in the horse spleen and bacteria ferritin by promoting the displacement of Fe(III) by Fe(II) in the ferroxidase center and accelerating the ferrous redox activity on the surface of the iron core[37,56,57]. In addition, Watt et al. have found that ferritins from horse spleen and *A. vinelandii* possessed different redox potentials. Compared with the horse spleen ferritin (190 mV, −310 mV, and −416 mV at pH 7.0, 8.0, and 9.0, respectively), the *A. vinelandii* ferritin exhibits a more negative and stable reduction potential (−420 mV) at above pHs, indicative of stronger iron retaining ability by the *A. vinelandii* ferritin. In addition, during the reduction of horse spleen ferritin, there are almost 2 $H^+$ transferred to the core for each $Fe^{3+}$ reduced to the $Fe^{2+}$, which was absent in the bacteria ferritin[35]. This phenomenon corresponds well to our DFT result that the iron core in HFn owned stronger hydron proton binding ability compared with the archaeal ones[35,53]. These results demonstrated that phosphate affects the formation process as well as the reaction ability of the iron core.

More importantly, our disclosing of the natural ferritin SOD nanozyme and catalytic mechanism would provide a new platform for biomedical applications. Superoxide could not only directly damage the enzyme with Fe-S center, like the aconitase and succinate dehydrogenase, but also transform into other highly toxic reactive oxygen radicals such as hydroxyl radicals and peroxynitrite, which are harmful to most cells and tissues. So superoxide plays an important role in many diseases, including radiation damage, stroke, neurodegenerative diseases, cancer, and so on[58]. Previous research has applied the natural superoxide dismutase to treat rheumatoid arthritis[59], osteoarthritis[60] and mitigate the radiotherapy and chemotherapy damage[61]. However, owing to the low cellular uptake, immunogenicity, and short half-life, their clinical translation was greatly restricted[58]. With the high stability and easy modification of ferritin structures, the finding of the SOD-like activity of natural ferritin nanozyme would not only contribute to the understanding of the biominerals but also offer a new strategy for the superoxide-related disease. For example, our previous works have proved that HFn itself could target the tumor cells with high specificity and sensitivity based on their TfR1 binding ability[62]. In addition, the overexpression of superoxide dismutase has been found to inhibit the growth or invasion of the tumor cells[63,64]. Combined with these two characteristics, it was believed the ferritin nanozyme could be used to enhance tumor therapy. Moreover, the ferritin cages could not only be easily modified to endow them with different targeting characteristics but also act as a good drug carrier[65], which further broadened their application scenarios. Besides tumor therapy, the ferritin nanozyme

might be also used as an antioxidant for both in vitro and in vivo applications (e.g., anti-inflammation, and anti-aging). Nevertheless, our finding here just preliminarily resolved the catalytic mechanism of the ferritin nanozyme, much work was needed to further optimize their structures and activities before their applications.

Furthermore, it is worth noting that we only examined the difference and mechanism of SOD-like activity of the ferritins. It is likely that other enzyme-like activities of natural ferritin iron core may also have important physiological and evolutionary functions that are interesting and remain to be further explored. In addition, ferritin only constituted a few parts of the natural nanozymes. Much more work is needed to discover the existence of various natural nanozymes and explore their physiological functions. Hopefully, our studies on the enzyme-like activity of ferritins may pave the way for more systematic investigations.

## Methods
### Materials
All chemicals were of analytical grade. NaCl, Yeast extract, peptone, $(NH_4)_2FeSO_4$, $Fe(NO_3)_3 \cdot 9H_2O$, $FeCl_3$, NaOH, $FePO_4$, $Fe_2O_3$, hydrogen peroxide ($H_2O_2$, 30 wt%), 3, 3', 5, 5'-Tetramethylbenzidine (TMB), 2',7'-dichlorofluorescein (DCFH-DA), ampicillin, kanamycin, spectinomycin and isopropyl-β-D-thiogalactoside (IPTG) were purchased from Sigma-Aldrich (USA). Fe foil was purchased from the Alfa Aesar. Human CuZnSOD was purchased from Absin (China), SOD assay kit, 5,5-Dimethyl-1-pyrroline-N-oxide (DMPO), and Microbial viability assay kit were purchased from DOJINDO (Japan). LIVE/DEAD BacLight Bacterial Viability Kit was purchased from Thermo Fisher (USA).

### Multiple sequence alignment and phylogenetic tree construction
The amino acid sequences of the DNA binding protein from starved *Escherichia coli* str. K-12 substr. MG1655 (EcDps, NP_415333.1), DNA protection protein from *Sulfolobus solfataricus* (ssDps, NP_343470.1), Bacterioferritin from *Escherichia coli* str. K-12 substr. MG1655 (EcBfr, NP_417795.1), Non-heme ferritin from Enterobacteriaceae (EcFTn, WP_000917208.1), heavy chain ferritin from *Homo sapiens* (HFn, NP_002023.2), light chain ferritin from *Homo sapiens* (LFn, NP_000137.2), ferritin from *Pyrococcus furiosus* (pfFn, WP_011011871.1), ferritin from *Pyrococcus yayanosii* (pyFn, WP_013905435.1), ferritin from *Caenorhabditis elegans* (CeFn, NP_504944.2), ferritin from *Crassostrea gigas* (CgFn, NP_001292267.1), ferritin from *Siniperca chuatsi* (XP_044062311), heavy chain ferritin from *Xenopus laevis* (NP_001084057.1), heavy chain ferritin from *Anolis carolinensis* (XP_003221882), heavy chain ferritin from *Gopherus evgoodei* (XP_030416668), heavy chain ferritin from *Coturnix japonica* (XP_015719168), bacterioferritin from *Parabacteroides goldsteinii* strain BFG-241 (WP_007659302), bacterioferritin from *Alistipes onderdonkii* subsp. vulgaris strain 3BBH6 (WP_022332956.1), bacterioferritin from *Thermus scotoductus* (WP_038069286.1), bacterioferritin from *Streptococcus equinus* (WP_074533576.1), bacterioferritin from *Leptospira weilii* (WP_002621798), bacterioferritin from *Pseudomonas aeruginosa* (PaBfr, WP_003092078.1), ferritin from *Pseudomonas aeruginosa* (PaFTn, WP_003093668.1), ferritin from *Thermus scotoductus* (TsFn, WP_038069286.1), ferritin from *Mycobacterium tuberculosis* (MtFn, NP_218358.1), ferritin from *Candidatus Korarchaeum* (WP_012309233.1), ferritin from *Staphylothermus marinus* (WP_011839058), ferritin from *Thermosphaera aggregans* (TaFn, WP_013129153) and ferritin from *Thermococcus barophilus* (TbFn, WP_013467695) were obtained from the NCBI. The multiple sequence alignment was conducted by Cluster W and the sequence similarity was marked by Genedoc software (the color referred to amino acid similarity, red: 100%, blue: 80%, gray: 60%), and the Mega5 was applied to construct the phylogenetic tree by the neighbor-joining method, bootstrapped with 1000 replicates.

## Plasmid preparation, wild type and biomineralized ferritin expression and purification

Total 18 different ferritins were selected for the purification and enzyme-like activity test, including EcDps, ssDps, EcFTn, EcBfr, HFn, LFn, pfFn, pyFn, CeFn, CgFn, PaBfr, PaFTn, TsFn, MtFn, TaFn, TbFn, human mitochondrial ferritin (MFn) and human H/L heteropolymer ferritin (H/LFn). Except for the H/LFn, other 17 ferritin coding sequences were cloned to the pet22b plasmid with the NdeI and BamHI restriction site. The H/LFn expression vector was constructed by cloning the two coding sequences into petDuet-1 plasmid through NcoI/BamHI and NdeI/XhoI. The constructed plasmids were transformed to the BL21(DE3) competent cells to express the proteins. The transformed competent cells were cultured overnight in LB medium with $100 \mu g \, mL^{-1}$ ampicillin. The protein expression was induced by adding 1 mM isopropyl-β-D-thiogalactoside (IPTG, Sigma) when the OD600 reached 0.6−0.8, then the induced bacteria were further cultured for 8 h at 30 °C. The biomineralized ferritins were synthesized by adding freshly prepared ferrous salt into the bacteria culture medium just before the IPTG so that the excess iron could be biomineralized in the ferritin cavity during the ferritin expression. The induced bacteria were harvested by $3300 \times g$ centrifugation for 20 min and the pellets were resuspended for protein purification. The purification processes were referred to previous publications[29-34], except that all the phosphate salt buffer was substituted for the Tris-base or Hepes buffer.

## Characterization of the wild type and biosynthetic ferritin from different species

TEM: the ferritin samples (10 μL, 0.1 mg mL⁻¹) were embedded in a Plasma Cleaner HPDC32G-treated copper grid and stained with 1% uranyl acetate for 1 min, then imaged with a JEM-1400 80 kV TEM (JEOL, Japan). DLS: the average diameter of the ferritin samples (20 μL, 0.1 mg mL⁻¹) was analyzed by the DynaPro Titan with a temperature-controlled micro-sampler of 25 °C (Wyatt Technology, U.S.A.). Inductively coupled plasma mass spectrometry (ICP-MS): the element contents of ferritin samples from different species were analyzed by the Agilent ICPMS7800.

## Ferritin iron core removal and protein shell digestion

The iron core removal method was referred to previous publication[66]: the biosynthetic ferritins were added to the 0.1 M sodium acetate buffer containing 1% thioglycolic acid, pH 5.5 and placed in 4 °C for 18 h, then excess 2-2′-bipyridyl was added to chelate the ferrous. The apo-ferritin was then acquired by the G75 desalting column (GE Healthcare). The protein shell was digested by the protease K: protease K was added to the ferritin solution in 20 mM Tris-base buffer pH 8.0 and placed at 55 °C, 24 h for complete digestion of the ferritin shell.

## Synthesis of ferritin reconstituted with different iron/phosphate ratio

The reconstitution experiment was modified from previous publication[67]: ferritins were placed in the 50 mM MES buffer, pH 6.5, then the freshly prepared ammonium ferrous sulfate solution was added to the ferritin solution in the frequency of 500/ferritin every 10 min at the room temperature for the ferrous oxidant and nucleation. Ferritins with different iron/phosphate ratios were synthesized by adding relative sodium hydrogen phosphate before the ferrous salt. After the reaction, the mixture was centrifuged at $13,800 \times g$ for 10 min to remove the excess sediment. The free ferric ions were then removed by the G75 desalting column.

## Enzyme-like activity test

Superoxide dismutase-like activity test: the SOD-like activities of the ferritins and ferrihydrites were evaluated by the SOD assay kit (Dojindo, Japan), which was based on the Xanthine-Xanthine oxidase system. The xanthine oxidase could oxidase the substrate xanthine leading to the reduced xanthine oxidase, which subsequently oxidase the oxygen to generate the superoxide. The superoxide could react with the water-soluble tetrazolium salt (WST-1) in the assay kit to generate the WST-1 formazan dye, which exhibited a specific absorption peak at 450 nm, so the superoxide level correlated with OD 450 absorbance. The SOD agent could diminish the superoxide, leading to the decrease of the absorbance at 450 nm. Therefore, SOD activity was defined as the "inhibition rate (%)" of the superoxide's reaction with WST-1, and calculated as [(Ablank 1 - Ablank 3) − (Asample - Ablank 2)]/ (Ablank 1 - Ablank 3) x 100, where the Ablank 1 referred to OD450 of above reaction system without SOD agents, Ablank 3 referred to reaction system without the SOD agent and xanthine oxidase, Asample referred to reaction system with SOD agents, Ablank 2 referred to reaction system with SOD agents but without the xanthine oxidase. During the actual test, 20 μL samples with specific protein concentrations determined by the BCA assay or iron content determined by ICP-MS were mixed with 200 μL WST-1 working solution, pH 7.4, then 20 μL xanthine oxidase was added to start the reaction and the mixture was placed at 37 °C for 30 min, then OD450 was monitored by the microplate reader. One unit of SOD was defined as the amount of the enzyme in 20 μL of the sample solution that inhibited the reduction reaction of WST-1 with superoxide anion by 50%. Likewise, the SOD units base on the commercial CuZnSOD was calculated based on the IC50 of commercial SOD and its relative SOD units.

Peroxidase-like and oxidase-like activity test: the peroxidase-like and oxidase-like activity was evaluated by using TMB (dissolved in DMSO, 10 mg mL⁻¹) as the substrate. Briefly, 190 μL mixture containing 1 μL TMB (final concentration 0.1 mg mL⁻¹) and 5 μL $H_2O_2$ (final concentration 0.5 M) in the 0.2 M NaAc buffer, pH 4.5 was added to the 10 μL ferritin or ferrihydrite samples (final concentration 0.1 mg mL⁻¹). The activity was evaluated by monitoring the absorbance of the oxTMB at 652 nm in a 30 min time course. The oxidase-like activity was performed likewise except that the $H_2O_2$ was absent in the mixture.

Catalase-like activity test: catalase-like activity was determined by the specific oxygen electrode on a multi-parameter analyzer (JPSJ-606L, Leici China). The ferritin or ferrihydrite was mixed with the $H_2O_2$ aqueous solution of different concentrations in the 20 mM Hepes buffer, pH 7.4, the final concentration of the ferritin or ferrihydrite was 2.5 μg mL⁻¹, and the oxygen generation rate was recorded for 2 min.

## Urate generation Assay of the Xanthine oxidase system

The urate generation assay was conducted by referring to previous literatures[68-71]. Briefly, xanthine oxidase (30 μL of 0.77 U mL⁻¹ xanthine oxidase) was added to a solution of 20 mM Hepes buffer, pH 7.4, containing xanthine (150 μmol L⁻¹) at a final volume of 3.0 mL at 25 °C. Urate production was monitored by the change of absorbance at 290 nm. For the ferritin groups, the ferritins were added before xanthin oxidase in the final concentration of 100 μg mL⁻¹.

## Modification and purification of the three-fold axis mutated human heavy chain ferritin

Amino acids at the 3-fold axis of HFn were substituted for the relative amino acids in the *Pseudomonas aeruginosa* Bfr (T123E, C131R, E135K, T136D). The modified HFn mutant sequence was cloned to the pet22b plasmid, and the HFn mutant was purified as the wild type.

## Synthesis of phosphate doped ferrihydrite

Phosphate doped 2-line or 6-line ferrihydrite was synthesized by the method modified from previous publication[72]. All the solutions were prepared by the ddH₂O with a resistivity of 18 MΩ · cm. 2-line ferrihydrite and phosphate-doped 2-line ferrihydrite: 40 g Fe(NO₃)₃·9H₂O were added to 500 mL distilled water or those containing relative K₂HPO₄ (Pi/Fe = 0.1, 0.2, 0.5, 1) under vigorously stirring, then 1 M NaOH was added to the solution with continuous stirring to adjust the pH to 7−8. Then the ferrihydrites were centrifuged at $5000 \times g$ for

20 min to precipitate the ferrihydrites and washed with distilled water 5 times to remove the electrolyte. The precipitations were then freeze-fried for storage or further use. 6-line ferrihydrite and phosphate doped 6-line ferrihydrite: the distilled water containing no or 0.215 g, 0.172 g, 1.075 g, 2.15 g $K_2HPO_4$ (Pi/Fe = 0.1, 0.2, 0.5, 1) were placed in the 75 °C oven for pre-heating, then 5 g $Fe(NO_3)_3 \cdot 9H_2O$ was added to the pre-heated solution under stirring and the mixture was placed back to the oven for 10−12 min. After the reaction, the solution was transferred to the ice for rapid cooling. The cooled solutions were then dialyzed against distilled water for 7 days, the dialyzed water was replaced 3 times every day. The dialyzed materials were freeze-fried for storage or further use.

## Characterization of the ferritin iron core and the phosphate-doped ferrihydrite

High resolution transmission electron microscope (HRTEM, FEI Tecnai G2 F30, USA) with the selected area electron diffraction (SAED) was used to characterize the morphology and crystal lattice structure of the ferritin iron core and the phosphate-doped ferrihydrites. X-ray photoelectron spectroscopy (XPS, Thermo escalab 250Xi, monochromatic AlKα radiation accelerating voltage of 14.8 kV, power of 150 W, USA) was applied to characterize the element constitution and valence state. FTIR spectrum (Nicolet IS10, USA) was used to determine the specific functional group by scanning in the range of 400−4000 $cm^{-1}$ in the step of 4 $cm^{-1}$. X-ray powder diffractometer (XRD, Brucker D8 advance, Germany) was used to characterize the crystal structure of the ferritins and phosphate-dope ferrihydrite with the Cu (Kα) radiation in the range of 10−90° with the step of 4° $min^{-1}$.

## Surface phosphate absorption of the ferritin iron core and the ferrihydrite

The ferrihydrite surface absorption experiment was modified from the previous publication[72]. The freeze-dried 2-line ferrihydrite was suspended in deionized water containing 0.1 M NaCl by sonication. The iron content of the prepared ferrihydrite solution was determined by the ICP-MS, and then phosphate salt of relative molar (Pi/Fe molar ratio = 0.1, 0.2, 1) was added to the ferrihydrite solution. The mixture was stirred for 24 h for sufficient absorption and to avoid potential precipitation. Then the samples were centrifuged at 12,000 × g for 10 min to separate the phosphate-absorbed ferrihydrite and washed three times with $ddH_2O$. The precipitation was freeze-fried for determining the iron and phosphate content and tested for SOD-like activities. Ferritin iron core surface absorption experiment was performed likewise. Firstly the iron-reconstituted ferritin was synthesized by adding the ammonium ferrous sulfate to the different ferritin in 50 mM MES buffer, pH 6.5, and further purified by the G75 desalting column. Then the phosphate solution of relative molar concentration was added to the reconstituted ferritin and the mixtures were further incubated at 4 °C for 24 h, then the samples were centrifuged at 12,000 × g for 10 min and purified by G75 desalting column. The protein concentration was detected by the BCA protein assay kit and SOD-like activity was tested.

## Synchrotron radiation characterization of the ferritin iron core

The iron chemical state and coordination environment were assessed via X-ray absorption fine structure (XAFS) using Fe foil, $Fe_2O_3$, and $FePO_4$ as reference materials. Biomineralized ferritin samples underwent a dialysis process against distilled water to eliminate salts. Subsequently, the dialyzed solutions were freeze-dried, thoroughly ground, and compressed into uniform pellets affixed to 3 M tape. Fe-K edge XAFS spectra were acquired at beamline 1W1B of the Beijing Synchrotron Radiation Facility (BSRF) in China, employing the transmission mode for sample and reference measurements. Normalized XANES data were analyzed via least-squares fitting using IFEFFIT Athena software from CARS (Consortium for Advanced Radiation

Sources at the University of Chicago) to ascertain the ratio of iron species. For EXAFS data analysis, IFEFFIT Artemis software (also from CARS) was utilized, employing the Fe (Im-3m) phase for data fitting purposes.

## DFT studies of SOD-like activity of phosphate ferrihydrite

The slab model of the (001) surface of ferrihydrite was employed to investigate its catalytic activities for the disproportionation of superoxide. The most stable P-doped ferrihydrite was constructed by replacing one Fe atom with a P atom. Utilizing these slab models, DFT calculations were conducted through the Vienna Ab initio Simulation Package (VASP 5.4.4)[73]. The calculations employed the Perdew-Burke-Ernzerhof functional, augmented with Grimme's semi-empirical "D3BJ" dispersion term (PBE-D3BJ)[74]. To describe valence electrons, plane wave basis sets with an energy cutoff of 450 eV were utilized, while projector-augmented-wave pseudopotentials described core electrons. Convergence criteria were set at $10^{-5}$ eV for energy and 0.02 eV $Å^{-1}$ for atomic force. Brillouin zone sampling was achieved using $3 \times 3 \times 1$ Monkhorst-Pack[75] k-point meshes in all calculations.

## Intermediate ferrous detection and the recycle activity test

The intermediate ferrous ions detection was performed according to the method modified from the previous method[76]. Briefly, the ferritins or the phosphate-doped ferrihydrite were mixed with xanthine and ferrozine in the same total iron content of 44.8 $\mu g\,mL^{-1}$ in the 20 mM Hepes buffer, pH 7.4. Then xanthine oxidase was added to start the reaction and the generation of ferrous was monitored by the $Fe^{2+}$-Ferrozine that exhibited specific absorbance at 562 nm. The Ferrous standard curve was acquired by adding gradient ammonium ferrous sulfate to ferrozine. The standard curve was calculated by the linear regression method. The recycle activity experiment was performed by reacting ferritins or phosphate-doped ferrihydrite with gradient xanthine for 24 h in the presence of XOD. Then the archaeal ferritin groups were heated at 90 °C for 15 min to denature the xanthine oxidase, and ferritins were separated from the mixture by the centrifugation and desalting column. The phosphate-doped ferrihydrite was separated by centrifugation and $ddH_2O$ washing. Then the separated ferritins and phosphate-doped ferrihydrites were used for further SOD-like activity test.

## Electron spin resonance

The electron spin resonance spectroscopy (ESR) was used to examine the superoxide scavenging ability of the ferritins: xanthine oxidase (XOD) and xanthine was used to generate superoxide in the 20 mM Hepes buffer, pH 7.4, 100 mM DMPO was used to trap the superoxide by forming the DMPO/OOH·. The reaction was started by the addition of the XOD.

## The antioxidant experiments of ferritin transgenic E. coli

Spot assay: The sensitivity to the superoxide generator paraquat of transgenic bacteria was evaluated by the spot assay. Transformed strains with empty plasmid or reconstituted expression vectors were cultured in the fresh LB medium. When the OD600 reached 0.8, the inoculum was induced by 1 mM IPTG at 30 °C for 8 h, then these strains were 10 times gradient diluted and spotted on the solid LB medium containing 0.2 mM paraquat to analyze their sensitivity.

Growth curve monitor: The transformed BL21(DE3) bacteria with empty plasmid or the reconstituted expression vectors were cultured in the fresh LB medium containing 100 $\mu g\,mL^{-1}$ ampicillin until the OD600 reached 0.6−0.8, then the inoculum was diluted to the same $OD_{600}$ by the fresh LB medium. Subsequently, 1 mM IPTG and 1 mM paraquat were simultaneously added to the medium when the OD600 reached 0.2 and the inoculums were further incubated at 37 °C, 200 rpm for 12 h. The OD600 absorbance was recorded every 2 h.

Reactive oxygen species (ROS) assay: the intracellular ROS level was determined by the 2′-7′-dichlorofluorescein diacetate (DCFH-DA). DCFH-DA could be taken up by cells where cellular esterase cleaves off the acetyl groups, resulting in the DCFH. The cellular-generated ROS could oxidize the DCFH to DCF, which emits green fluorescence with an excitation wavelength of 488 nm and an emission wavelength of 525 nm. The fluorescence levels were evaluated by Flow Cytometry. Flow Cytometry could assign a channel to the fluorescence cell according to their fluorescence intensity. With the increase of cellular fluorescence intensity, the cells would be distributed in higher channels (corresponding to a shift to a higher X-axis range). During the actual assay, transformed bacteria were treated as before. After the treatment, the bacteria were harvested and washed with PBS buffer, pH 7.4. Then DCFH-DA was added to the resuspend bacteria in the final concentration of 10 μM and incubated at 37 °C for 30 min, then the bacteria were washed twice by PBS, pH 7.4 and detected by Flow Cytometry.

Bacteria viability assay: The bacterial viability was checked by colony formation experiment. Briefly, the bacteria containing different plasmid was treated as mentioned above. After the paraquat treatment, the transgenic bacteria were plated with proper dilution and cultured at 37 °C overnight, then the bacterial numbers were calculated as CFU mL$^{-1}$.

Bacteria dehydrogenase activity assay: The microbial viability assay kit (Dojindo) was used to detect the bacteria viability through their dehydrogenase activity. The bacteria containing different plasmid was treated as mentioned above. After treatment, 190 μL bacteria inoculum was transferred to the 96-well plates and 10 μL working solution (containing WST-8 and mediator) was added to each well. Then the plate was placed at 37 °C for 2 h–4 h and OD450 was recorded by SpectraMax. Besides, the living/Dead Baclight bacteria viability kit was used to determine the living (SYTO-9, FITC) and dead (PI, Texas Red) bacteria by the confocal laser scanning microscopy (CLSM) according to the supplier's manuscript.

Morphology observation: Scanning electron microscopy was applied to observe the transformed bacteria morphology. The paraquat-treated bacteria were harvested and washed with PBS buffer, and the precipitate was fixed by 2.5% glutaraldehyde at 4 °C for 8 h. Then the fixed sample was washed three times with the saline and further dehydrated by a series of ethanol/water mixture. The sample was then dried by the Automatic critical point dryer (CPD300) and observed by the scanning electron microscopy (SU8010).

## Gene knockout and substitution method by CRISPR/Cas9 in BL21(DE3)

The *Bfr* gene knockout and substitution were conducted according to Yang et al.[77], and gRNA for *Bfr* was designed by the CRISPOR website. The pecCas9 plasmid containing λ-RED and Cas9 gene was transformed into the BL21(DE3) strain, and positive colonies were selected to induce the expression of λ-RED and make competent cells. Next, the pecgRNA plasmid and homologous fragment of *Bfr* upstream and downstream with or without *HFn* insertion were electroporated into the competent cells. The individual colonies were verified by colony PCR and DNA sequencing. Finally, the verified colonies were used for pecgRNA and pecCas plasmids curing by adding the rhamnose and sucrose respectively. All the plasmids and primers used in this study were listed in Supplementary Table 2.

## Reporting summary

Further information on research design is available in the Nature Portfolio Reporting Summary linked to this article.

## Data availability

The data supporting the findings of this study are available from the corresponding authors upon request. All the PDB Structures are available on the PDB database (https://www.rcsb.org/). Source data for the figures and supplementary figures are provided as a Source Data file. Source data are provided with this paper.

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

## Acknowledgements
We thank Prof. Yaling Wang and Beijing Synchrotron Radiation Facility (BSRF), China for the technical support for the X-ray absorption fine structure. We thank Dr. Yujing Ren for his technical help in the XAFS data analysis. Supported by the National Natural Science Foundation of China (82122037, K.F., 81930050, X.Y., 22121003, X.Y.), National Key Research and Development Program of China (No. 2021YFC2102900, K.F.), CAS Project for Young Scientists in Basic Research (YSBR-089, K.F.), CAS Interdisciplinary Innovation Team (JCTD-2020-08, K.F.), and Key Laboratory of Biomacromolecules, Chinese Academy of Sciences (O3CCSDZ301, K.F.).

## Author contributions
K.F. conceived and designed the whole project. L.M. designed and performed the experiments. J.J.Z., X.F.G. designed and performed the DFT calculation. L.M., N.Z. designed and purified the HFn protein mutant. R.Z., L.F. contributed to protein purification and data analysis. L.Y. contributed to the manuscript writing. Y.C. supervised the XAFS. K.F., L.M., and X.Y. wrote the manuscript with input from all authors.

## Competing interests
The authors declare no competing interests.
