## [Peer Review File · Nature Communications]

Reviewers' Comments:

Reviewer #1:

Remarks to the Author:

In this work, the authors report that biogenic ferritins act as natural SOD enzymes and find that the biogenic iron cores of prokaryotic ferritins have a higher ability to scavenge superoxide radicals than that of eukaryotes. In addition, it is revealed that the protein structure of ferritins determines the P/Fe ratio in the core and that different P/Fe content leads to different iron nucleus lattice structures and SOD-like activities. This work uses extensive methodology and provides lots of experimental data that can support their conclusions. Overall, it is a very interesting work.

However, in my opinion, there are some issues that should be clarified and modified.

1. The authors demonstrated that the SOD-like catalysis of ferritins is different from the Haber-Weiss reaction involving Fe ions, is it possible to show the specific process of the SOD-like reaction of ferritins by using the reaction equations or schematic diagram? Also, does ferritin produce hydroxyl radicals in exerting SOD-like activity? Please provide ESR data.

2. There are inconsistencies in the results of SOD-like activity measurements of some ferritins in the paper. For example, the "inhibition rate of SOD" of "pfFn" in Fig 1f was about 40%, however, in Fig S3a, the "inhibition rate of SOD" of "pfFn" was less than 20%; The data of "pyFn" in Fig S3a and Fig S3c were also inconsistent; In Fig S3e-g, the ferritins with "0 mM Fe content" corresponds to the ferritins purified from the basal medium, but their "inhibition rate of SOD" are not consistent with the results in Fig S3a; The OD562 of "Pi/Fe=1" in Fig S15d was lower than that of "Pi/Fe=0.2 and 0.5". Please explain these discrepancies.

3. What are the pH conditions when measuring SOD-like and CAT activity? Please add in the experimental methods.

4. Please define the meaning of the abbreviations used in the text/figures where appropriate, such as apo-pfFn, -Fe, Fe-in vitro, Fe-in vivo, HFn-WT, HFn-3-Fold-M, HFn-3-Fold-M-Fe, etc.

5. Some figures in the text were cited in reverse order, e.g. Supplementary Figure 1b and 1c, d; Fig. 4d and 4e; Supplementary Figure 12f and 12g, h.

6. "However, the formation of H₂O₂ after adsorption of the second HOO• on ferrihydrite is endothermic with an increase in energy by 0.73 eV (Fig. 5h)": Please check if 0.73 eV is correct.

7. There are some typing errors in this article, such as "pffn" in Fig 2f, "3-pore" should be "3-Fold" in Supplementary Figure 7c and 7d.

8. The article referred to Supplementary Figure 16f, which is not available in the SI.

9. Supplementary Figure 2c: The unit of oxygen generation rate should be mg·L⁻¹·min⁻¹.

10. Supplementary Figure 4: What are the red, blue, and gray represent respectively?

11. Supplementary Figure 5a: "CkFn" is not mentioned in this article; "Thermococcus barophilus (TaFn)" should be "Thermococcus barophilus (TbFn)"; "TbFn" is misspelled as "TaFn" in Supplementary Figure 5b.

12. Please revise the format of the references carefully, noting issues such as letter case, subscripts, and page numbers.

Reviewer #2:

Remarks to the Author:

Fan and coworkers have described mechanism behind antioxidant role of ferritins with high SOD like activity. They established a relationship between the strength of catalytic activity and the source of species. The results are incremental to the previous studies in the field, and most of the results in the manuscript are in agreement with what authors have claimed. I have listed some of my concerns below-:

1. The authors should express the bacterial viability in terms of colony forming units/mL instead of OD450.

2. The TEM images need to be cleaner showing a higher magnification image in inset.

3. Authors need to demonstrate the potential applications of their findings in order to make this study suitable for broad readership of Nature Communications. One application may be enhanced tumor therapy.

Overall, the manuscript is well written and should be published in Nature Communications after major revision.

Reviewer #3:

Remarks to the Author:

This work on the SOD activity of ferritin is remarkable, both for the themes addressed, the angles taken and the subjects explored. The text is very well written and very convincing. I think it should be accepted in Nature Comm., but some of the experiments need to be better described. I give a list of what could be improved. I think this is key to the potential impact of the article. That is why I have indicated that these are "major revisions".

(a) From the outset, I wondered whether the observed activity might be associated with metal ion leakage from the inorganic core and ferritin, especially as it has been evidenced that this occurs under oxidative stress. Some answers are given at the end, but I think they should be mentioned from the start (around line 127, perhaps just to indicate that further experiments will be provided at the end to ensure that the activity is not associated with the metal ion released from ferritin).

(b) My most important concern is the description of SOD-type activity.

First of all, the assay they use is based on Xanthine-Xanthine Oxidase (X/XO) with a UV-vis marker. This should be made clearer than simply referring to an assay kit with the name of a supplier. I had to download the supplier's data sheet to better understand what had been done. It is technically important to indicate (i) the nature of the UV-vis marker (WST-1?), (ii) the concentration used for the marker (iii) the pH.

The SOD like activity is reported as a « % of inhibition of SOD » in the figures and with the exp. part line 578-593, but this label is incorrect: the assay does not measure an inhibition of SOD at all. Here how it goes: the enzyme xanthine oxidase (XOox) oxidizes the substrate xanthine leading to XOred and, when a proper pH is used, XOred re-oxidizes using dioxygen and leading to a flow of ca. 75 % of superoxide (and some H₂O₂). This superoxide reacts with the UV-vis marker (here WST-1) and changes its color. When a SOD like agent is added to the medium, it competes with WST-1 and limits its reaction by superoxide (in a way, there is less superoxide available for WST-1 as it is partially used by the SOD-like agent). So, what is measured here is an "inhibition of the marker's reaction with superoxide".

SOD-like activity is indicated as "% SOD inhibition" in the figures and on line 578-593 of the exp. section, but this label is misleading, as the assay does not measure SOD inhibition at all. Here's how it works: the oxidized form of the enzyme xanthine oxidase (XOox) oxidizes the substrate xanthine leading to XOred and, when an appropriate pH is used, XOred re-oxidizes using dioxygen and leading to a flux of around 75% superoxide (and some H₂O₂). This superoxide reacts with the UV-vis marker (here WST-1) and changes color. When a SOD-like agent is added to the medium, it competes with WST-1 and limits its reaction with superoxide (in a way, there is less superoxide available for WST-1 as it is partially utilized by the SOD-like agent). What is being measured here, therefore, is "inhibition of the marker's reaction with superoxide".

As this is a competition between WST-1 and the SOD-agent, the measured "inhibition" will be dependent of the concentration of WST-1 and of the SOD-agent.

They state "the sample with equal quality or equal iron content was mixed with 200 µL of WST-1 working solution". Can they provide this "equal quantity and equal iron content" and WST-1 concentration more explicitly? What is the actual metal content? I assume it was measured by ICP-MS: can they specify that? Also, from other figures/exp (see fig6), it looks like they can provide a ferritin amount in µg. Could this be used here? I mean indicate the amount of iron AND the µg/mL of ferritin used in the assay?

Another concern is there is here no way to evaluate whether the activity is a good one or not. It should be possible by the kinetic method and a kcat could be recalculated provided the kcat of the marker is known. That would be a lot a work to reperform the analyses using the kinetic method for this paper, but I would advise to try and set-up the kinetic experiment in the future. With this set of experiments, what I would suggest is to measure the activity of a commercially available protein SOD. An idea could be to estimate a number of SOD units that would provide an inhibition of the WST-1 oxidation in a range similar to what they have obtained for the ferritin. There would be two ways to report the results: (a) give a result for a certain amount of Fe and just say it is eq. to ca. xx SOD unit (b) report the results per quantity of metal ion content, but I guess this would be less convincing as SOD contain one metal ion and ferritin a large core in which most probably

only a few are catalytically active (those onto the surface and probably not those in the bulk of the particle?).

SOD activities are not given in such a way that the experiment can be redone if we wish, or compared with another experiment. How the so-called "SOD inhibition %" is calculated is unclear and does not exactly match the supplier's description (see the equation, SOD activity (inhibition rate %) = [(Ablank 1 - Ablank 3) - (Asample - Ablank 2)] / (Ablank 1 - Ablank 3) x 100, which is different from that given in the exp. part of the article). Could they describe more precisely what they have done? Note that this measurement should be sensitive to incubation time and that the most robust way to perform this type of measurement is the kinetic method also described by the supplier.

Another problem is that there is no way of assessing whether the activity is good or not. This should be possible using the kinetic method, and a k_{cat} could be recalculated provided that the k_{cat} of the marker is known. Although it surely would be a lot of work to redo the analyses using the kinetic method for the present article, I would advise trying to set up the kinetic experiment in the future. With this series of experiments, I would suggest measuring the activity of a commercially available SOD protein. One idea might be to find a number of SOD units that would provide inhibition of WST-1 oxidation in a range similar to that obtained for ferritin. There would be two ways of presenting the results: (a) give a result for a certain amount of Fe and simply say that it is equivalent to about xx SOD units (b) present the results by amount of metal ions, but I suppose this would be less convincing as SOD contains one metal ion and ferritin a large core in which, most likely, only a few are catalytically active (those on the surface and probably not those in the bulk of the particle?).

(c) Another concern is that this SOD-assay can give false positive results. This would be the case for instance if the SOD-like agent inhibited the enzyme XO. In this case, the reduction in superoxide flow will not be due by the reaction of the agent with superoxide BUT to the reduction in superoxide production by XO. This can be checked by measuring, upon addition of the putative SOD agent, the activity of XO through the oxidation of xanthine into urate. This can be checked by UV-visible (see (1) C. Policar, in *Redox Act. Ther.* (Eds.: I. Batinić-Haberle, J.S. Rebouças, I. Spasojević), Humana Press, Published By Springer Nature, 2016, pp. 125–164, see box p. 154-156 ; (2) S. Durot, C. Policar, F. Cisnetti, F. Lambert, J.-P. Renault, G. Pelosi, G. Blain, H. Korri-Yousoufi, J.-P. Mahy, *Eur. J. Inorg. Chem.* 2005, 2005, 3513–3523 ; (3) Weiss RH, Flickinger AG, Rivers WJ, Hardy MM, Aston KW, Ryan US, Riley DP. Evaluation of activity of putative superoxide dismutase mimics. Direct analysis by stopped-flow kinetics. *J Biol Chem.* 1993;268(31):23049–54 ; (4) Faulkner KM, Liochev SI, Fridovich I. Stable Mn(III) porphyrins mimic SOD in vitro and substitute for it in vivo. *J Biol Chem.* 1994;269(38):23471–6.)

(d) Paragraph Protein structure of ferritins determining the iron/phosphate content in the core:
-I like this idea that the protein core shapes the iron core. But can they be more specific and may be give a reference?
-how do they prepare the "biomineralized ferritin"? I could not find any information about that, in the text, exp part or SI.

(e) Paragraphe "Phosphate coordinated with the iron atom leading to high SOD-like activit"
-They write: "It has been previously shown that phosphate exists on either the surface of the iron core in horse spleen ferritin or in the interior of the bacteria ferritin, which suggests that the phosphate inner the iron core might be the determining factor of the SOD-like activity." I do not see how this information from the literature suggest that. This is what follows that show it. Can they rephrase?
-They indicate (line 311) that the iron oxidation state is (+III) ; I agree, but this is not necessarily its actual charge (due to delocalization onto the ligand). May be rephrase.
-Pre-edge: please provide a reference for the pre-edge indicating an Oh structure for Fe(III). Similarly, identification of FePO₄ by EXAFS, please, give a preference for the comparison with a EXAFS spectrum for FePO₄.

(f) Figure 5: I guess route (i) has more phosphate than (h). Can that they be indicated more clearly in the figure/text?

(g) Figure 6g/h: (g) the ESR spectrum is not that of superoxide but that of superoxide trapped by

DMPO. Please, change the caption. The amount indicated on the figure as xx $\mu\text{g}/\text{mL}$: are they in μg of enzyme or of iron? It seems it is Fe (from h) but specify that more clearly on the figure. (h) Can they specify which line(s) was used to estimate the intensity of the ESR?

(h) The results with DCF are not very clear to me in the processing. Can they be more explicit about that? Do they look at the area below the curve, at the maximum absorbance? Sorry, this is just that I am not used to this kind of reporting, but I guess other readers will be like me.

Overall, this an excellent paper but that needs to be made more precise on the overall chemical and physico-chemical description. I hope my comments will help improving the impact of this nice study within the community of chemists.

Bibliography: the references are not homogeneously provided (titles in capital letters or not for instance).

Response to Reviewer 1:

Reviewer #1 (Remarks to the Author):

In this work, the authors report that biogenic ferritins act as natural SOD enzymes and find that the biogenic iron cores of prokaryotic ferritins have a higher ability to scavenge superoxide radicals than that of eukaryotes. In addition, it is revealed that the protein structure of ferritins determines the P/Fe ratio in the core and that different P/Fe content leads to different iron nucleus lattice structures and SOD-like activities. This work uses extensive methodology and provides lots of experimental data that can support their conclusions. Overall, it is a very interesting work. However, in my opinion, there are some issues that should be clarified and modified.

Response: We really appreciate the reviewer's careful review and positive comments on our manuscript, we have addressed the issues according to the reviewer's suggestions.

1. The authors demonstrated that the SOD-like catalysis of ferritins is different from the Haber-Weiss reaction involving Fe ions, is it possible to show the specific process of the SOD-like reaction of ferritins by using the reaction equations or schematic diagram?

Response: As suggested, we have drawn schematic diagrams of the ferritin with different cores by DFT (Fig 5g & h), in which the ferrihydrite and P-doped ferrihydrite were used to mimic the ferritin core with lower and higher P/Fe ratios, respectively. As shown in Figure 5g-i, the detailed superoxide diminishing processes include the absorption and disproportionation of HOO^\bullet on the ferrihydrite core surface. In the ferritin with a lower P/Fe ratio, the first HOO^\bullet is absorbed on the ferrihydrite surface where the O-H splits without energy barriers. This process causes the O_2 production and retains an H atom on the ferrihydrite surface. Subsequently, the second HOO^\bullet comes into the active site and experiences the same O-H split process. Due to its high affinity to the H proton, the ferrihydrite would form the stable H-binding steady state and could not catalyze the dismutase reaction efficiently (Fig 5h). Contrarily, on the ferritin core with a higher P/Fe ratio, the doped phosphate could apparently lower the affinity of ferrihydrite to H, which would lower the energy barrier of the second HOO^\bullet transforming to H_2O_2 and facilitate the regeneration of the active surface (Fig 5i).

Figure 5. DFT analysis of the reaction process of Ferrihydrite and superoxide. (g) Proposed two possible reaction pathways for superoxide with products of H_2O_2 and O_2 (main reaction) or O_2 and H^* (side reaction) and catalytic centers in Ferrihydrite (side reaction) and P-doped Ferrihydrite (main reaction). (h) Reaction profiles with key intermediate structures and reaction energies (eV) for Ferrihydrite. (i) Reaction profiles with key intermediate structures and reaction energies (eV) for P-doped Ferrihydrite. Only important structural fragments were shown for clarity.

Also, does ferritin produce hydroxyl radicals in exerting SOD-like activity? Please provide ESR data.

Response: Thanks very much for the expert question. Both in the Electron Spin Resonance experiments of ferritins and ferric ions, we used the DMPO as the trapper for the free radicals, which could efficiently trap O-, C-, N-, and S-centered radicals to form distinguishable ESR spectra¹⁻⁴. As shown in Fig 6g and Supplementary 15h-j, with the substrate concentration increasing, the superoxide produced by the xanthine-xanthine oxidase system was apparently transformed into hydroxyl radicals by ferric ions, however, there is no hydroxyl radical production in the ferritin groups with the same iron content. These data indicate that the ferritins diminished the superoxide without the hydroxyl radicals forming.

Fig 6. (g) ESR spectrum of the DMPO/ HOO• in the presence of XOD/Xan and gradient pyFn, the marked concentration referred to the iron content of ferritins. (h) The ESR intensity of DMPO/ HOO• of HFn, pfFn and pyFn groups. The line at 3488.125 G of DMPO/ HOO• spectra was used to represent the intensity.

Supplementary Figure 15 (h, i) ESR spectrum of the DMPO/ HOO• generated by Xan/XOD in the presence of gradient HFn and pfFn, marked concentration referred to the iron content of ferritins determined by ICP/MS. (j) ESR spectra of DMPO/ HOO• generated by Xan/XOD in the presence of gradient Fe³⁺.

2. There are inconsistencies in the results of SOD-like activity measurements of some ferritins in the paper. For example, the "inhibition rate of SOD" of "pfFn" in Fig 1f was about 40%, however, in Fig S3a, the "inhibition rate of SOD" of "pfFn" was less than 20%; The data of "pyFn" in Fig S3a and Fig S3c were also inconsistent; In Fig S3e-g, the ferritins with "0 mM Fe content" corresponds to the ferritins purified from the basal medium, but their "inhibition rate of SOD" are not consistent with the results in Fig S3a;

Response: We appreciate the insightful comments. These variations likely resulted from different preparations of the proteins for each of the experiments. To address these discrepancies, we have performed the above SOD activity tests again with indicated concentrations of ferritins (Revised Fig 1f, Revised Supplementary Fig 3c

& f-h). The inhibition rates for different ferritins were consistent with our previous results dose-dependently.

Revised Figure 1. (f) The SOD-like activity of apo-pfFn (iron-removed ferritin), pfFn, and biom mineralized pfFn (pfFn-Fe).

Revised Supplementary Figure 3. (f-h) SOD-like activity of biom mineralized ferritins with gradient iron content.

The OD562 of "Pi/Fe=1" in Fig S15d was lower than that of "Pi/Fe=0.2 and 0.5". Please explain these discrepancies.

Response: We have noticed the discrepancies and confirmed with additional experiments (Figure only for reviewer 1). Of note, with similar SOD-like activities and superoxide diminishing efficiency in the same iron content, pyFn also exhibited apparently lower OD562 than the pfFn group. We believe that the OD562 only reflected the ferrous ions generation during the reaction of ferritin or ferrihydrite with superoxide, not the SOD activities. This discrepancy in the ferrous ion production and SOD activities also reflects that the ferritin and ferrihydrite decompose the superoxide in a manner different from that of the ferric ion.

Figure only for Reviewer 1. The ferrous generation curve of different phosphate-doped ferrihydrite in the iron concentration at 44.8 $\mu\text{g/mL}$. All data are expressed as means \pm S.D.

3. What are the pH conditions when measuring SOD-like and CAT activity? Please add in the experimental methods.

Response: We apologize for missing the information. The pH conditions during the SOD-like and CAT-like activities were both pH 7.4, which has been added to the experiment methods of the revised manuscript (page 27, paragraph 3 & Page 28, paragraph 2).

4. Please define the meaning of the abbreviations used in the text/figures where appropriate, such as apo-pfFn, -Fe, Fe-in vitro, Fe-in vivo, HF_n-WT, HF_n-3-Fold-M, HF_n-3-Fold-M-Fe, etc.

Response: Thank you very much for pointing out these issues. We have defined the meaning of the abbreviations used in the revised manuscript. They are added in Figure 1f legend, Figure 2e & f legend, Page 10, line 19-20, and Supplementary Figure 7b legend, respectively.

5. Some figures in the text were cited in reverse order, e.g. Supplementary Figure 1b and 1c, d; Fig. 4d and 4e; Supplementary Figure 12f and 12g, h.

Response: Thank you for pointing out these problems. We have checked the citations of all the figures and revised their orders accordingly. Please refer to Page 5, line 3-5 & Page 13 line 27, and Page 14 in the revised version.

6. “However, the formation of H₂O₂ after adsorption of the second HOO• on ferrihydrite is endothermic with an increase in energy by 0.73 eV (Fig. 5h)”: Please check if 0.73 eV is correct.

Response: We apologize for the mistake and have revised the change of reaction energies to 0.7 eV, please refer to Page 16, line 26.

7. There are some typing errors in this article, such as "pFfn" in Fig 2f, "3-pore" should be "3-Fold" in Supplementary Figure 7c and 7d.

Response: We apologize for the mistake and we have made corrections in the revised manuscript (Figure 2f and Supplementary Figure 7c & d).

8. The article referred to Supplementary Figure 16f, which is not available in the SI.

Response: We apologize for the mistake and have corrected it in the revised manuscript.

9. Supplementary Figure 2c: The unit of oxygen generation rate should be $\text{mg}\cdot\text{L}^{-1}\cdot\text{min}^{-1}$.

Response: Thank you for the comment. We have corrected the mistake in the revised manuscript in Supplementary Figure 2c & f.

Revised Supplementary Figure 2. The peroxidase, oxidase, and catalase-like activities of ferritins from different species. (a) Peroxidase-like activity of ferritins purified from the basic medium. (b) Oxidase-like activity of ferritins purified from the basic medium. (c) Catalase-like activity of ferritins purified from the basic medium. (d) Peroxidase-like activity of ferritins purified from medium containing excess iron. (e) Oxidase-like activity of ferritins purified from medium containing excess iron. (f) Catalase-like activity of ferritins purified from medium containing excess iron. All data are means \pm S.D

10. Supplementary Figure 4: What are the red, blue, and gray represent respectively?

Response: We apologize for missing the information. The colors marked in Supplementary Figure 4 referred

to the similarity of amino acid sequences, red: 100%, blue: 80%, and gray: 60%. We have added the information in the Supplementary Figure 4 legend.

11. Supplementary Figure 5a: "CkFn" is not mentioned in this article; "Thermococcus barophilus (TaFn)" should be "Thermococcus barophilus (TbFn)"; "TbFn" is misspelled as "TaFn" in Supplementary Figure 5b.

Response: We apologize for the mistakes. CkFn has been removed from the "Supplementary Figure 5a" and corrected TaFn to TbFn in the "Supplementary Figure 5a" legend.

12. Please revise the format of the references carefully, noting issues such as letter case, subscripts, and page numbers.

Response: Thank you very much for the suggestion, we have checked the references carefully and unified the format.

Response to reviewer 2:

Reviewer #2 (Remarks to the Author):

Fan and coworkers have described mechanism behind antioxidant role of ferritins with high SOD like activity. They established a relationship between the strength of catalytic activity and the source of species. The results are incremental to the previous studies in the field, and most of the results in the manuscript are in agreement with what authors have claimed. I have listed some of my concerns below-:

1. The authors should express the bacterial viability in terms of colony forming units/mL instead of OD450.

Response: Thank you very much for the suggestion. Accordingly, colony-forming experiments were conducted after different strains were treated with the paraquat (Figure 7d).

Figure 7d. The colony formation of the different ferritin transgenic *E. coli* with or without the paraquat stimulation. All data are means \pm S.D, One-way ANOVA with Tukey's test, * $p < 0.05$, $n=3$.

2. The TEM images need to be cleaner showing a higher magnification image in inset.

Response: Thank you very much for the suggestion. We have provided TEM pictures with magnified images in the inset (Figure only for Reviewer 2, the collection of revised Figure 1d, Figure 3e, h, and Supplementary Figure 1b, Supplementary Figure 5c).

Figure only for Reviewer 2. TEM images of ferritins from different organisms. (a) Stained and unstained TEM images of biom mineralized ferritins from 8 different species, (b) Stained and unstained TEM images of ferritins of 8 different species purified from the basic medium. (c) Stained TEM images of ferritins from different species. (d) Stained TEM image of three-fold pore mutated HFn. (e) Stained and unstained TEM images of human heavy chain and light chain heteromeric ferritin and human mitochondrial ferritin. The scale bar referred to 50 nm, inset scale bar referred to 20 nm.

3. Authors need to demonstrate the potential applications of their findings in order to make this study suitable for broad readership of Nature Communications. One application may be enhanced tumor therapy.

Response: Thank you very much for the instrumental suggestion. While this manuscript focuses on the characterization and mechanisms of the ability of natural ferritin nanozymes to scavenge superoxide radicals, our data also indicate that the activity is physiologically important. The potential applications of the activity are what we are actively exploring in the laboratory at this moment. Therefore, we discussed the potential

biomedical applications of our findings in the Discussion part below (please also refer to Page 23-24 in the revised manuscript).

Discussion: More importantly, our disclosing of the natural ferritin SOD nanozyme and catalytic mechanism would provide a new platform for biomedical applications. Superoxide could not only directly damage the enzyme with Fe-S center, like the aconitase and succinate dehydrogenase, but also transform into other highly toxic reactive oxygen radicals such as hydroxyl radicals and peroxynitrite, which are harmful to most cells and tissues. So superoxide plays an important role in many diseases, including radiation damage, stroke, neurodegenerative diseases, cancer, and so on⁵⁸. Previous research has applied the natural superoxide dismutase to treat rheumatoid arthritis⁵⁹, osteoarthritis⁶⁰ and mitigate the radiotherapy and chemotherapy damage⁶¹; however, owing to the low cellular uptake, immunogenicity, and short half-life, their clinical translation was greatly restricted⁵⁸. With the high stability and easy modification of ferritin structures, the finding of the SOD-like activity of natural ferritin nanozyme would not only contribute to the understanding of the biominerals but also offer a new strategy for the superoxide-related disease. For example, our previous works have proved that HF_n itself could target the tumor cells with high specificity and sensitivity based on their TfR-1 binding ability⁶². Besides, the overexpression of superoxide dismutase has been found to inhibit the growth or invasion of the tumor cells^{63, 64}. Combined with these two characteristics, it was believed the ferritin nanozyme could be used to enhance tumor therapy. Moreover, the protein cages could not only be easily modified to endow them with different targeting characteristics but also act as a good drug carrier⁶⁵, which further broadened their application scenarios. Besides tumor therapy, the ferritin nanozyme might be also used as an antioxidant for both *in vitro* and *in vivo* applications (e.g. anti-inflammation, anti-aging). Nevertheless, our finding here just preliminarily resolved the catalytic mechanism of the ferritin nanozyme, much work was needed to further optimize their structures and activities before their application.

Overall, the manuscript is well written and should be published in Nature Communications after major revision.

Response: Thank you very much for the positive comments.

Response to reviewer 3:

Reviewer #3 (Remarks to the Author):

This work on the SOD activity of ferritin is remarkable, both for the themes addressed, the angles taken and the subjects explored. The text is very well written and very convincing. I think it should be accepted in Nature Comm., but some of the experiments need to be better described. I give a list of what could be improved. I think this is key to the potential impact of the article. That is why I have indicated that these are "major revisions".

Response: We really appreciate the positive comments and many detailed suggestions of the reviewer which were very helpful for us to refine our manuscript. We have carefully studied the review and revised our manuscript accordingly.

(a) From the outset, I wondered whether the observed activity might be associated with metal ion leakage from the inorganic core and ferritin, especially as it has been evidenced that this occurs under oxidative stress. Some answers are given at the end, but I think they should be mentioned from the start (around line 127, perhaps just to indicate that further experiments will be provided at the end to ensure that the activity is not associated with the metal ion released from ferritin).

Response: Thank you very much for the constructive suggestion. Metal leakage was also the main concern when we conducted this study. Therefore, we designed different experiments to prove the real activity sources from different aspects, including comparing the activity in the same iron content and examining the difference with the ferric-induced reaction. We have added the description about the metal ion around Page 5, line 25 in the revised manuscript according to the suggestion.

(b) My most important concern is the description of SOD-type activity.

First of all, the assay they use is based on Xanthine-Xanthine Oxidase (X/XO) with a UV-vis marker. This should be made clearer than simply referring to an assay kit with the name of a supplier. I had to download the supplier's data sheet to better understand what had been done. It is technically important to indicate (i) the nature of the UV-vis marker (WST-1?), (ii) the concentration used for the marker (iii) the pH.

Response: Thank you very much for the critical question and we apologize for not providing a detailed activity assay procedure. The detailed information of the SOD activity test has been provided in the Method section, including the assay principle, the experiment pH, and the inhibition rate calculation process. Please refer to Page 27-28 in our manuscript. However, the WST-1 was added according to the manual, the specific working concentration was not afforded to us though we have consulted the supplier.

The SOD like activity is reported as a « % of inhibition of SOD » in the figures and with the exp. part line 578-593, but this label is incorrect: the assay does not measure an inhibition of SOD at all. Here how it goes: the enzyme xanthine oxidase (XOox) oxidizes the substrate xanthine leading to XOred and, when a proper pH is used, XOred re-oxidizes using dioxygen and leading to a flow of ca. 75 % of superoxide (and some H₂O₂). This superoxide reacts with the UV-vis marker (here WST-1) and changes its color. When a SOD like agent is added to the medium, it competes with WST-1 and limits its reaction by superoxide (in a way, there is less superoxide available for WST-1 as it is partially used by the SOD-like agent). So, what is measured here is an “inhibition of the marker’s reaction with superoxide”.

SOD-like activity is indicated as "% SOD inhibition" in the figures and on line 578-593 of the exp. section, but this label is misleading, as the assay does not measure SOD inhibition at all. Here's how it works: the oxidized form of the enzyme xanthine oxidase (XOox) oxidizes the substrate xanthine leading to XOred and, when an appropriate pH is used, XOred re-oxidizes using dioxygen and leading to a flux of around 75% superoxide (and some H₂O₂). This superoxide reacts with the UV-vis marker (here WST-1) and changes color. When a SOD-like agent is added to the medium, it competes with WST-1 and limits its reaction with superoxide (in a way, there is less superoxide available for WST-1 as it is partially utilized by the SOD-like agent). What is being measured here, therefore, is "inhibition of the marker's reaction with superoxide".

As this is a competition between WST-1 and the SOD-agent, the measured “inhibition” will be dependent of the concentration of WST-1 and of the SOD-agent.

Response: We are grateful for the insightful comments and suggestions, and apologize for the confusion. We have changed all the Y-axis labels about SOD activity to “inhibition rate (%)” and described the detailed meaning of this label in the Method section.

They state "the sample with equal quality or equal iron content was mixed with 200 µL of WST-1 working solution". Can they provide this "equal quantity and equal iron content" and WST-1 concentration more explicitly? What is the actual metal content? I assume it was measured by ICP-MS: can they specify that? Also, from other figures/exp (see fig6), it looks like they can provide a ferritin amount in µg. Could this be used here? I mean indicate the amount of iron AND the µg/mL of ferritin used in the assay?

Response: We apologize for missing the information and appreciate the questions and comments. We have included the protein and iron quantification methods in the revised manuscript and provided the specific final ferritin and iron concentrations used in the Figures and legends.

Another concern is there is here no way to evaluate whether the activity is a good one or not. It should be possible by the kinetic method and a kcat could be recalculated provided the kcat of the marker is known.

That would be a lot a work to reperform the analyses using the kinetic method for this paper, but I would advise to try and set-up the kinetic experiment in the future. With this set of experiments, what I would suggest is to measure the activity of a commercially available protein SOD. An idea could be to estimate a number of SOD units that would provide an inhibition of the WST-1 oxidation in a range similar to what they have obtained for the ferritin. There would be two ways to report the results: (a) give a result for a certain amount of Fe and just say it is eq. to ca. xx SOD unit (b) report the results per quantity of metal ion content, but I guess this would be less convincing as SOD contain one metal ion and ferritin a large core in which most probably only a few are catalytically active (those onto the surface and probably not those in the bulk of the particle?).

Response: We are grateful for the expert and insightful comments and suggestions. We will set up the kinetic method of SOD activity in near future. As suggested, we measured the relative inhibition rate of commercial human CuZnSOD. As the inhibition rate might exist a little variation, we calculated the SOD units through IC50 of inhibition rate based on the commercial SOD units or the enzyme unit calculation method offered by the supplier. As shown in the Figure for Reviewer 3, the calculated SOD units reveal a similar tendency, illustrating the reliability of the results. (Please also refer to the revised Supplementary Figure 7m-n).

Figure only for Reviewer 3. The relative calculated SOD units of the ferritin nanozymes. (a) Inhibition rate on the reaction between the superoxide and WST-1 of commercial Human CuZnSOD. (b) Calculated SOD units of reconstituted HFn-Fe, H/LFn-Fe, MFn-Fe, pfFn-fe, pyFn-Fe based on the method of SOD assay kit or the commercial CuZnSOD. All data are means ± S.D.

SOD activities are not given in such a way that the experiment can be redone if we wish, or compared with another experiment. How the so-called "SOD inhibition %" is calculated is unclear and does not exactly match the supplier's description (see the equation, SOD activity (inhibition rate %) = [(Ablank 1 - Ablank 3) - (Asample - Ablank 2)] / (Ablank 1 - Ablank 3) x 100, which is different from that given in the exp. part of the article). Could they describe more precisely what they have done? Note that this measurement should be sensitive to incubation time and that the most robust way to perform this type of measurement is the kinetic method also described by the supplier.

Response: Thanks very much for pointing out the important issue. The SOD activity assays were carried out by following the supplier's instructions strictly and calculated the inhibition rate as $\frac{(A_{\text{blank 1}} - A_{\text{blank 3}}) - (A_{\text{sample}} - A_{\text{blank 2}})}{(A_{\text{blank 1}} - A_{\text{blank 3}})} \times 100$. However, as the "Ablank 2" and "Ablank 3" referred to the base values of the sample and blank groups, we simplified the presentation form of the calculation process as $(A_{\text{blank}} - A_{\text{sample}}) / A_{\text{blank}} \times 100$. We apologize for causing the confusion, and we have corrected the description in the experiment section. Furthermore, to verify the relationship between the incubation time and the SOD activities, we have determined the inhibition rate of HF_n, H/LF_n, MF_n, pf_n and py_n in the indicated time. As shown in the Figure below, there is no apparent differences of inhibition rates in the time range of 30 min to 6 h.

Figure only for reviewer 4. Time stability test of reconstituted ferritins' SOD activities. (a) HF_n. (b) H/LF_n. (c) MF_n. (d) pf_n. (e) py_n. All data are means \pm S.D.

Another problem is that there is no way of assessing whether the activity is good or not. This should be possible using the kinetic method, and a k_{cat} could be recalculated provided that the k_{cat} of the marker is known. Although it surely would be a lot of work to redo the analyses using the kinetic method for the present article, I would advise trying to set up the kinetic experiment in the future. With this series of experiments, I would suggest measuring the activity of a commercially available SOD protein. One idea might be to find a number of SOD units that would provide inhibition of WST-1 oxidation in a range similar to that obtained for ferritin. There would be two ways of presenting the results: (a) give a result for a certain amount of Fe and simply say

that it is equivalent to about xx SOD units (b) present the results by amount of metal ions, but I suppose this would be less convincing as SOD contains one metal ion and ferritin a large core in which, most likely, only a few are catalytically active (those on the surface and probably not those in the bulk of the particle?).

Response: Thanks very much for the question of the reviewer. To further address the concern about the SOD activities, we have conducted all the SOD activity assays of the ferritins in a gradient concentration manner, in which the iron and ferritin concentrations were expressed directly. (Figure only for Reviewer 5).

Figure only for Reviewer 5. SOD activities of ferritins. (a) The SOD-like activity of pfFn, biomineralized ferritins (pfFn-Fe) and biomineralized pfFn with the iron removal (apo-pfFn). (b) The SOD-like activity of pfFn, biomineralized pyFn (pyFn-Fe) and biomineralized pyFn with the iron removal (apo-pyFn). (c) The SOD-like activity of HFn biomineralized with gradient iron content. (d) The SOD-like activity of pfFn biomineralized with gradient iron content. (e) The SOD-like activity of pyFn biomineralized with gradient iron content. (f) The SOD-like activity of the reconstituted and biomineralized HFn and HFn purified from the basic medium. (g) The SOD-like activity of the reconstituted and biomineralized pfFn and pfFn purified from the basic medium. (h) The SOD-like activity of the HFn reconstituted with gradient phosphate/iron ratio. (i) The SOD-like activity of the pfFn reconstituted with gradient phosphate/iron ratio. (j) The SOD-like activity of the pyFn reconstituted with gradient phosphate/iron ratio. (k) SOD-like activities of reconstituted HFn, pfFn and pyFn in the same total iron content. (l) SOD-like activities of biomineralized HFn, pfFn and pyFn in the same total iron content. (m) SOD-like activity of biomineralized human HFn, MFn, H/LFn and the archaeal pfFn, pyFn. (n) SOD-like activity of reconstituted human HFn, MFn, H/LFn and the archaeal pfFn, pyFn. (o) SOD-like activity of reconstituted human HFn, MFn, H/LFn and the archaeal pfFn, pyFn in the same total iron content. (p) SOD-like activity of the phosphate-doped 2-line ferrihydrite in the same quality. (q) SOD-like activity of the phosphate-doped 2-line ferrihydrite in the same total iron content. (r) SOD-like activity of the phosphate-doped 6-line ferrihydrite in the same quality. (s) SOD-like activity of reconstituted HFn with gradient surface

binding phosphate. **(t)** SOD-like activity of the 2-line ferrihydrite with gradient surface binding phosphate. **(u)** SOD-like activity of reconstituted H/LFn with gradient surface binding phosphate. **(v)** SOD-like activity of reconstituted pfFn with gradient surface binding phosphate. **(w)** SOD-like activity of reconstituted pyFn with gradient surface binding phosphate. All data are means \pm S.D. Two-way ANOVA with Tukey's test, **** $p < 0.0001$, *** $p < 0.001$, ** $p < 0.01$, * $p < 0.05$, $n=3$.

(c) Another concern is that this SOD-assay can give false positive results. This would be the case for instance if the SOD-like agent inhibited the enzyme XO. In this case, the reduction in superoxide flow will not be due by the reaction of the agent with superoxide BUT to the reduction in superoxide production by XO. This can be checked by measuring, upon addition of the putative SOD agent, the activity of XO through the oxidation of xanthine into urate. This can be checked by UV-visible (see (1) C. Policar, in Redox Act. Ther. (Eds.: I. Batinić-Haberle, J.S. Rebouças, I. Spasojević), Humana Press, Published By Springer Nature, 2016, pp. 125–164, see box p. 154-156 ; (2) S. Durot, C. Policar, F. Cisnetti, F. Lambert, J.-P. Renault, G. Pelosi, G. Blain, H. Korri-Youssoufi, J.-P. Mahy, Eur. J. Inorg. Chem. 2005, 2005, 3513–3523 ; (3) Weiss RH, Flickinger AG, Rivers WJ, Hardy MM, Aston KW, Ryan US, Riley DP. Evaluation of activity of putative superoxide dismutase mimics. Direct analysis by stopped-flow kinetics. J Biol Chem. 1993;268(31):23049–54 ; (4) Faulkner KM, Liochev SI, Fridovich I. Stable Mn(III) porphyrins mimic SOD in vitro and substitute for it in vivo. J Biol Chem. 1994;269(38):23471–6.)

Response: Thanks for the very constructive suggestion. We have detected urate production by pfFn and pyFn groups that possess the highest SOD activities. As shown in the Figure only for Reviewer 6, these ferritins had similar urate production rate with the Ctrl group, indicating the SOD activities were not caused by the inhibition of xanthine oxidase. Please also refer to Supplementary Figure 3i & j in the revised Supporting information.

Figure only for Reviewer 6. The urate generation in the reaction of archaea ferritins, xanthine and xanthine oxidase. (a) the OD 290 change in the first 3 min of biomineralized pyFn or pyFn reacted with xanthine and xanthine oxidase, final ferritin concentration: 100 $\mu\text{g/mL}$. **(b)** The slope of OD 290 change of Ctrl, biomineralized pfFn and pyFn groups. All data are means \pm S.D. One-way ANOVA with Tukey's test, $n=3$.

(d) Paragraph Protein structure of ferritins determining the iron/phosphate content in the core:

-I like this idea that the protein core shapes the iron core. But can they be more specific and may be give a reference?

Response: Thanks very much for the positive comments of the reviewer. As we found that though synthesizing in the same condition (either by biomineralization or reconstitution), ferritin from humans all exhibited lower SOD activities and lower P/Fe ratio, so we implied that the ferritin structures could affect the core constitution (or the phosphate intake). Interestingly, previous work about the bacterioferritin structure has found that the anion existed in the entrance of the phosphate-soaked *E.coli* bacterioferritin⁵ and in the middle of the three-fold axis in the Fe-soaked bacterioferritin⁶. More importantly, when Fe-soaked bacterioferritin was soaked in the ferrous solution again, this sulfate would appear in the interior cavity of bacterioferritin, illustrating the anion might enter the ferritin cavity through the 3-fold axis with the help of ferrous intake. Furthermore, compared with HFn, the three-fold axis of bacterioferritin exhibited more hydrophobic, polar amino acids⁷ and relatively less negative electrostatic potential, which could facilitate the anion intake. We thus verified this assumption by the mutant experiment and systematical comparison of 40 different known ferritins from eukaryotes and prokaryotes. Overall, based on previous work, our results confirmed that the three-fold axis of ferritins could affect the phosphate entrance, cause different core constitutions and subsequent SOD activity. However, in this part, we focused on the effect of the ferritin structures. As seen in the reconstitution experiment in Figure 2, the synthesizing condition would also affect the specific core compositions, the final appearances of different ferritins depend on both factors. We have included more detailed information in this paragraph and modified the title to make it clear. Please also refer to Page 10, paragraph 1 in the revised manuscript.

-how do they prepare the “biomineralized ferritin”? I could not find any information about that, in the text, exp part or SI.

Response: Thanks for pointing out the question. We described the biomineralization process in Page 4, line 25-28 in the manuscript and the purification part in the exp. Part. We have added more detailed processes in Page 4 of the text and Page 26 of the “Material and Method”.

(e) Paragraph “Phosphate coordinated with the iron atom leading to high SOD-like activity”

-They write: “It has been previously shown that phosphate exists on either the surface of the iron core in horse spleen ferritin or in the interior of the bacteria ferritin, which suggests that the phosphate inner the iron core might be the determining factor of the SOD-like activity.” I do not see how this information from the literature

suggest that. This is what follows that show it. Can they rephrase?

Response: Thanks for pointing out the question. We have rephrased this description as “It has been previously shown that phosphate exists on either the surface of the iron core in horse spleen ferritin or in the interior of the bacteria ferritin, which suggests the different locations of the phosphate in the ferritin core.” Please also refer to Page 15, line 8 in the revised manuscript.

-They indicate (line 311) that the iron oxidation state is (+III) ; I agree, but this is not necessarily its actual charge (due to delocalization onto the ligand). May be rephrase.

Response: Thanks very much for the constructive suggestion. We have reorganized the description of the charge state of iron: “As shown by the X-ray absorption near edge structure (XANES) profiles, the absorption edges of all the ferritins were similar to the Fe_2O_3 and FePO_4 , indicating that the Fe in the ferritins exhibited similar electronic structure as these two referred materials”. Please also refer to Page 15, line 22 in the revised manuscript.

-Pre-edge: please provide a reference for the pre-edge indicating an Oh structure for Fe(III). Similarly, identification of FePO_4 by EXAFS, please, give a reference for the comparison with a EXAFS spectrum for FePO_4 .

Response: Thanks very much for the suggestion. We have added the relative references about the pre-edge analysis of the Fe(III)⁸ and EXAFS comparison with FePO_4 ⁹ in the revised version, please also refer to Page 15, line 24 & line 28 in the revised manuscript.

8. Westre, T. E. et al. A Multiplet Analysis of Fe K-Edge $1s \rightarrow 3d$ Pre-Edge Features of Iron Complexes. J. Am. Chem. Soc. 119, 6297–6314 (1997).

9. Orikasa, Y. et al. Noncrystalline Nanocomposites as a Remedy for the Low Diffusivity of Multivalent Ions in Battery Cathodes. Chem. Mater. 32, 1011–1021 (2020).

(f) Figure 5: I guess route (i) has more phosphate than (h). Can that they be indicated more clearly in the figure/text?

Response: Thanks for pointing out the issue. We have added specific descriptions of these two different items in the Figure 5 legend.

Figure 5. DFT analysis of reaction process of Ferrihydrite and superoxide. (g) Proposed two possible reaction pathways for superoxide with products of H_2O_2 and O_2 (main reaction) or O_2 and H^* (side reaction) and catalytic centers in Ferrihydrite and P-doped Ferrihydrite. (h) Reaction profiles with key intermediate structures and reaction energies (eV) for ferrihydrite. (i) Reaction profiles with key intermediate structures and reaction energies (eV) for P-doped ferrihydrite, only important structural fragments were shown for clarity.

(g) Figure 6g/h: (g) the ESR spectrum is not that of superoxide but that of superoxide trapped by DMPO. Please, change the caption. The amount indicated on the figure as xx $\mu\text{g}/\text{mL}$: are they in μg of enzyme or of iron? It seems it is Fe (from h) but specify that more clearly on the figure. (h) Can they specify which line(s) was used to estimate the intensity of the ESR?

Response: We apologize for these confusion and mistakes and appreciate the questions. We have corrected the caption about the ESR spectra in Figure 6 and Supplementary Figure 15h-j. Meanwhile, the meaning of the concentrations has been also added in the legends. Additionally, to quantify the ESR intensity, we chose the line at 3488.125 G to estimate the relative intensity of different groups, which has been also added in the Figure 6h legend.

Revised Fig 6. (g) ESR spectrum of the DMPO/ HOO• in the presence of XOD/Xan and gradient pyFn, the marked concentration referred to iron content. (h) The ESR intensity of DMPO/ HOO• of HFfn, pfFn and pyFn groups, the line at 3488.125 G of DMPO/ HOO• spectra was used to represent the intensity. (i) Relative activity (%) of pyFn-Fe and P-doped Ferrihydrite (Pi:Fe=1:1) in the presence of Xan/XOD.

Revised Supplementary Figure 15 (h. i) ESR spectrum of the DMPO/ HOO• generated by Xan/XOD in the presence of gradient HFfn and pfFn, marked concentration referred to the iron content of ferritins determined by ICP/MS. (j) ESR spectra of DMPO/ HOO• in the reaction of Xan/XOD in the presence of gradient Fe³⁺.

(h) The results with DCF are not very clear to me in the processing. Can they be more explicit about that? Do they look at the area below the curve, at the maximum absorbance? Sorry, this is just that I am not used to this kind of reporting, but I guess other readers will be like me.

Response: Thank you very much for the question. DCFH-DA is a reactive oxygen radical indicator, which could enter the cell and react with the ROS to form fluorescent DCF. The DCF has an excitation wavelength of 488 nm and an emission wavelength of 525 nm, which could be detected by Flow cytometry. In principle, when a fluorescent cell passes through the laser beam of the Flow cytometry, it will produce a peak or pulse of photon emission over time, which could be detected by a photomultiplier tube and converted the

fluorescence signal to a voltage pulse. Flow cytometry would assign a channel to the cell according to the pulse intensity. The more intense the fluorescence, the higher the channel number the cell event is assigned. The channels are usually viewed on a log scale on the X-axis. So, the fluorescence intensity could be justified by the shift of the events on the X-axis. With the increasing of fluorescence intensity, the cell events would be distributed on higher X-axis ranges of Flow cytometry histogram. We have also added related descriptions in Page 31-32 of the Method part in the revised manuscript.

Overall, this an excellent paper but that needs to be made more precise on the overall chemical and physico-chemical description. I hope my comments will help improving the impact of this nice study within the community of chemists.

Response: We really appreciate these expert and insightful questions and suggestions which have helped us greatly in improving our studies and preparing the manuscript.

Bibliography: the references are not homogeneously provided (titles in capital letters or not for instance).

Response: Thanks very much for pointing out the question. We have re-checked the references and unified the format of the references.

Reference:

- 1 Finkelstein, E., Rosen, G. M. & Rauckman, E. J. Spin trapping. Kinetics of the reaction of superoxide and hydroxyl radicals with nitrones. *Journal of the American Chemical Society* **102**, 4994-4999 (1980).
- 2 Motohashi, N. & Mori, I. Superoxide-dependent formation of hydroxyl radical catalyzed by transferrin. *FEBS Letters* **157**, 197-199, (1983).
- 3 Clément, J.-L. *et al.* Assignment of the EPR Spectrum of 5,5-Dimethyl-1-pyrroline N-Oxide (DMPO) Superoxide Spin Adduct. *The Journal of Organic Chemistry* **70**, 1198-1203, doi:10.1021/jo048518z (2005).
- 4 Lloyd, R. V., Hanna, P. M. & Mason, R. P. The Origin of the Hydroxyl Radical Oxygen in the Fenton Reaction. *Free Radical Biology and Medicine* **22**, 885-888, (1997).
- 5 Crow, A., Lawson, T. L., Lewin, A., Moore, G. R. & Brun, N. E. L. Structural Basis for Iron Mineralization by Bacterioferritin. *Journal of the American Chemical Society* **131**, 6808-6813, (2009).
- 6 Weeratunga, S. K. *et al.* Structural Studies of Bacterioferritin B from *Pseudomonas aeruginosa* Suggest a Gating Mechanism for Iron Uptake via the Ferroxidase Center. *Biochemistry* **49**, 1160-1175 (2010).
- 7 Aronovitz N, Neeman M, Zarivach R. Ferritin Iron Mineralization and Storage: From Structure to Function. In: Iron Oxides (Ed.) (2016).
- 8 Westre, T. E. *et al.* A Multiplet Analysis of Fe K-Edge 1s → 3d Pre-Edge Features of Iron Complexes. *Journal of the American Chemical Society* **119**, 6297-6314 (1997).
- 9 Orikasa, Y. *et al.* Noncrystalline Nanocomposites as a Remedy for the Low Diffusivity of Multivalent Ions in Battery Cathodes. *Chemistry of Materials* **32**, 1011-1021 (2020).

Reviewers' Comments:

Reviewer #1:

Remarks to the Author:

The author has carefully explained the issues I have raised and has revised the article accordingly. I think this paper can be accepted with some editorial changes.

Reviewer #2:

Remarks to the Author:

This manuscript by Fan and co-workers shows the SOD like activity of ferritins. The authors have done a commendable job in addressing the reviewer's comments. I recommend this manuscript to be published in Nature Communications.

Reviewer #3:

Remarks to the Author:

I think the authors did a good job with this second version that should be accepted.

Only one point still puzzles me, for the community and for their future work. Indeed, if the supplier of the SOD measurement kit does not provide the concentration in WST, this a difficulty for providing a kcat for comparison with other compounds from the literature. Most of us do not use a kit and that works perfectly while we can control the concentration in markor we put and thgius back-calculate the kcat.

Be careful in the added sections, the ref are given before the punctuation mark. Tehre are also dome places where liter is l and not L (capital letter).

Reviewer #1 (Remarks to the Author):

The author has carefully explained the issues I have raised and has revised the article accordingly. I think this paper can be accepted with some editorial changes.

Response: We really appreciate the reviewer's careful review and constructive suggestions that have helped us improve the clarity and accuracy of our manuscript.

Reviewer #2 (Remarks to the Author):

This manuscript by Fan and co-workers shows the SOD like activity of ferritins. The authors have done a commendable job in addressing the reviewer's comments. I recommend this manuscript to be published in Nature Communications.

Response:

Response: We are very grateful for the positive feedback provided by the reviewer.

Reviewer #3 (Remarks to the Author):

I think the authors did a good job with this second version that should be accepted. Only one point still puzzles me, for the community and for their future work. Indeed, if the supplier of the SOD measurement kit does not provide the concentration in WST, this a difficulty for providing a kcat for comparison with other compounds from the literature. Most of us do not use a kit and that works perfectly while we can control the concentration in markor we put and thgius back-calculate the kcat.

Response: We really appreciate the reviewer's positive comments on our manuscript. We agree well with the reviewer's opinion that the concentration of the Chromogenic substrate WST is very important for the research on the SOD, and we have negotiated with the supplier again. Fortunately, to help us better conduct the related SOD research in future, they provided the final concentration of WST [Redacted] for the kit we used. As suggested, we will carry out specific kinetics research in the near future.

([Redacted])

Be careful in the added sections, the ref are given before the punctuation mark. Tehre are also dome places where liter is l and not L (capital letter).

Response: Thanks to the reviewer for pointing out this issue, we have checked the reference and modified the format of the reference. The unit of l was modified to L in the Enzyme-like activity test of the Methods part.